# MLL3 regulates the *CDKN2A* tumor suppressor locus in liver cancer

Changyu Zhu[1†], Yadira M Soto-Feliciano[2,3*†], John P Morris[1,4†], Chun-Hao Huang[1], Richard P Koche[5], Yu-jui Ho[1], Ana Banito[1], Chun-Wei Chen[1], Aditya Shroff[1], Sha Tian[1], Geulah Livshits[1], Chi-Chao Chen[1], Myles Fennell[1], Scott A Armstrong[6], C David Allis[2], Darjus F Tschaharganeh[7*], Scott W Lowe[1,8*]

[1]Department of Cancer Biology and Genetics, Memorial Sloan Kettering Cancer Center, New York, United States; [2]Laboratory of Chromatin Biology and Epigenetics, The Rockefeller University, New York, United States; [3]Koch Institute for Integrative Cancer Research, Massachusetts Institute of Technology, Cambridge, United States; [4]Department of Pharmacology, The University of North Carolina at Chapel Hill, Chapel Hill, United States; [5]Center for Epigenetics Research, Memorial Sloan Kettering Cancer Center, New York, United States; [6]Dana-Farber Cancer Institute, Boston, United States; [7]Helmholtz-University Group "Cell Plasticity and Epigenetic Remodeling", German Cancer Research Center, Heidelberg, Germany; [8]Howard Hughes Medical Institute, New York, United States

*For correspondence:
ysoto@mit.edu (YMS-F);
d.tschaharganeh@dkfz.de (DFT);
lowes@mskcc.org (SWL)

†These authors contributed equally to this work

**Abstract** Mutations in genes encoding components of chromatin modifying and remodeling complexes are among the most frequently observed somatic events in human cancers. For example, missense and nonsense mutations targeting the mixed lineage leukemia family member 3 (MLL3, encoded by *KMT2C*) histone methyltransferase occur in a range of solid tumors, and heterozygous deletions encompassing *KMT2C* occur in a subset of aggressive leukemias. Although MLL3 loss can promote tumorigenesis in mice, the molecular targets and biological processes by which MLL3 suppresses tumorigenesis remain poorly characterized. Here, we combined genetic, epigenomic, and animal modeling approaches to demonstrate that one of the mechanisms by which MLL3 links chromatin remodeling to tumor suppression is by co-activating the *Cdkn2a* tumor suppressor locus. Disruption of *Kmt2c* cooperates with *Myc* overexpression in the development of murine hepatocellular carcinoma (HCC), in which MLL3 binding to the *Cdkn2a* locus is blunted, resulting in reduced H3K4 methylation and low expression levels of the locus-encoded tumor suppressors p16/Ink4a and p19/Arf. Conversely, elevated *KMT2C* expression increases its binding to the *CDKN2A* locus and co-activates gene transcription. Endogenous *Kmt2c* restoration reverses these chromatin and transcriptional effects and triggers Ink4a/Arf-dependent apoptosis. Underscoring the human relevance of this epistasis, we found that genomic alterations in *KMT2C* and *CDKN2A* were associated with similar transcriptional profiles in human HCC samples. These results collectively point to a new mechanism for disrupting *CDKN2A* activity during cancer development and, in doing so, link MLL3 to an established tumor suppressor network.

## Editor's evaluation

This paper convincingly shows that MLL3 regulates the CDKN2A tumor suppressor in MYC-driven liver cancers. The use of in vivo models and epigenomic analysis made the findings particularly robust. This work significantly advances our understanding of the function of MLL3 in cancer.

## Introduction

Hepatocellular carcinoma (HCC) is a deadly primary liver cancer with a 5 year survival rate of only 18% (*Jemal et al., 2017*). HCC is currently the fourth most frequent cause of cancer-related mortality worldwide, and its incidence continues to grow (*Llovet et al., 2021*). Genomic alterations found in HCC are highly diverse and are characterized by promoter mutations in *TERT* (telomerase reverse transcriptase), amplifications, or chromosomal gains encompassing the *MYC* oncogene, activating hotspot mutations in *CTNNB1* (β-catenin), and inactivating mutations and deletions in the *TP53* and *CDKN2A* tumor suppressor genes (2017; *Schulze et al., 2015*).

Among these alterations, genetic gain of *MYC* and inactivation of tumor suppressor p53 are known to cooperate to drive tumorigenesis in HCC (*Molina-Sánchez et al., 2020*). Mechanistically, oncogenic MYC activation triggers increased expression of the tumor suppressor ARF, one of two proteins encoded in *CDKN2A* in alternative reading frames. ARF binds to the E3 ubiquitin ligase MDM2 to prevent p53 degradation, leading to apoptosis to restrain MYC-driven tumorigenesis (*Lowe and Sherr, 2003*). However, it is unclear how the *CDKN2A* locus is regulated in response to MYC overexpression.

Beyond these well-studied drivers, HCC frequently harbors mutations in one or more chromatin modifying enzymes, including MLL3 (encoded by *KMT2C*; *Fujimoto et al., 2012*; *Kan et al., 2013*). MLL3 is a component of the COMPASS-like complex that has structural and functional similarities to the developmentally essential *Drosophila* Trithorax-related complex (*Schuettengruber et al., 2017*). This multiprotein complex controls gene expression through its histone H3 lysine 4 (H3K4) methyltransferase activity, which establishes chromatin modifications most often associated with transcriptional activation (*Shilatifard, 2012*). Most studies have shown that MLL3 and its paralog MLL4 (encoded by *KMT2D*) typically catalyze H3K4 monomethylation (H3K4me1) at enhancers (*Herz et al., 2012*; *Hu et al., 2013*), while the MLL1/2 complex is responsible for H3K4 trimethylation (H3K4me3) at promoters and enhancers in a locus-specific manner (*Denissov et al., 2014*; *Rickels et al., 2016*; *Wang et al., 2009*).

While less characterized, MLL3/4 regulation of promoter activity is emerging as an additional mechanism connecting the COMPASS-like complex to gene expression. Some publications report that H3K4me1 enrichment at promoters has been associated with gene repression (*Cheng et al., 2014*), and MLL3 inactivation decreases H3K4me3 levels at the promoters of metabolism-related genes in normal murine livers (*Valekunja et al., 2013*) and human liver cancer cells (*Ananthanarayanan et al., 2011*). Furthermore, a recent study in leukemia cells demonstrated that MLL3 and MLL4, in the absence of MLL 1/2 complex, are capable of binding to promoters to activate tumor suppressor genes (*Soto-Feliciano et al., 2023*). These divergent results suggest that the genomic binding pattern and functions of MLL3 are highly context dependent.

Notably, HCC also harbors mutations in *KMT2D* (*Cleary et al., 2013*), while KDM6A/UTX, an H3K27 demethylase within the COMPASS-like complex, has been functionally established as a potent tumor suppressor in pancreatic and liver cancers (*Revia et al., 2022*). These observations suggest that epigenetic-based mechanisms of gene regulation controlled by the MLL3 complex may constrain HCC development. However, because chromatin regulators such as ARID1A often exhibit context-specific tumor suppressive and oncogenic roles in liver cancer development (*Sun et al., 2017*), it is unclear whether MLL3 is a bona fide tumor suppressor in HCC. We therefore employed mouse models of HCC to investigate the molecular targets of MLL3 and the biological processes it affects.

## Results

### MLL3 is a tumor suppressor in *Myc*-driven liver cancer

To better understand the functional significance of genes commonly inactivated in HCC, including a number of chromatin regulators, we selected 12 genes with recurrent inactivating mutations in human HCC (*Cancer Genome Atlas Research Network, 2017*; *Ahn et al., 2014*; *Fujimoto et al., 2012*; *Figure 1—figure supplement 1A*) and performed a CRISPR-based in vivo screen to determine whether they behave as tumor suppressors in HCC. Specifically, the screen tested whether loss of each of these 12 genes would drive hepatic tumorigenesis in cooperation with *Myc*—one of the most frequently gained and/or amplified oncogenes in HCC (*Huang et al., 2014*). We applied hydrodynamic tail vein injection (HTVI) in wild-type mice to directly introduce genetic manipulations into adult

hepatocytes in vivo (*Bell et al., 2007*). We introduced both a transposon vector for stable genomic integration of oncogenic *Myc* cDNA and plasmids designed for transient expression of Cas9 and single guide RNAs (sgRNAs; a mix of two for each gene; *Figure 1B*; *Largaespada, 2009*; *Moon et al., 2019*; *Tschaharganeh et al., 2014*; *Xue et al., 2014*). 3 months after HTVI, only sg*Kmt2c* resulted in liver tumor formation with high penetrance (*Figure 1—figure supplement 1B*), suggesting that MLL3 likely acts as a tumor suppressor to constrain Myc-driven liver cancer. Supporting this idea, *KMT2C* mutations co-occur with *MYC* genomic gains and amplifications in human HCC tumors (*Figure 1A*).

To validate and extend the results from the screen, we applied the same approach to test whether the screen phenotype could be recapitulated with oncogenic *Myc* and single *Kmt2c*-targeted sgRNAs (*Figure 1B*). Mice injected with an *Myc* cDNA transposon combined with either of two independent Cas9/*Kmt2c* sgRNAs (*Myc*; sg*Kmt2c*.1 or *Myc*; sg*Kmt2c*.2) developed liver tumors, with a slightly later onset and slightly longer survival than mice receiving the *Myc* transposon combined with an sgRNA targeting *Trp53* (*Myc*; sg*Trp53*; *Figure 1C and D*). In contrast, mice injected with *Myc* and a control sgRNA (sg*Chrom8*) did not succumb to disease over the observation period (*Figure 1C*). These findings were confirmed in an independent cohort of mice (*Figure 1—figure supplement 2b*). Analyses of tumor-derived genomic DNA revealed insertions and deletions (indels) in either *Kmt2c* or *Trp53* depending on the genotype of tumor-derived cells (*Figure 1—figure supplement 2B*). DNA sequencing of the CRISPR-targeted region from two independent *Myc*; sg*Kmt2c* tumors revealed indels predicted to generate premature stop codons (*Figure 1—figure supplement 2C*). In one case, the indel was heterozygous, implying that even partial suppression of *Kmt2c* can promote tumorigenesis. In support of this, GFP-linked *Kmt2c* shRNAs efficiently cooperated with *Myc* overexpression to drive liver cancer, producing tumors with 50–80% reduction in *Kmt2c* mRNA expression (*Figure 1—figure supplement 2D–G*). sh*Kmt2c*.2 resulted in less potent knockdown than sh*Kmt2c*.1 yet produced faster tumor formation, suggesting that, as in acute myeloid leukemia (*Chen et al., 2014*), MLL3 can likely act as a haploinsufficient tumor suppressor in liver cancer (*Figure 1—figure supplement 2E, G*).

Apart from *MYC*, *CTNNB1* (β-catenin) is a frequently mutated oncogene in human HCC (*Rebouissou et al., 2016*), although the co-occurrence between *CTNNB1* and *KMT2C* mutations was not statistically significant (*Figure 1A*). To test whether MLL3 loss can cooperate with oncogenic *CTNNB1* to promote liver tumorigenesis, we performed analogous HTVI of a transposon vector expressing constitutively active β-catenin (*Ctnnb1*-N90; *Tward et al., 2007*) in combination with *Kmt2c*- or *Trp53*-targeted sgRNAs (*Figure 1—figure supplement 3A*). However, no tumor formation was observed in mice that received the *Kmt2c*-targeted sgRNAs (*Figure 1—figure supplement 3B*), indicating that unlike p53, the tumor-suppressive role of MLL3 is specific to the oncogene and contexts.

## MLL3 loss alters the chromatin landscape of liver cancer cells

MLL3 and MLL4 are histone methyltransferases that can deposit the H3K4 monomethylation mark at genomic enhancers and intergenic regions during organ development (*Hu et al., 2013*). However, more studies indicate that MLL3 and MLL4 are also capable of binding to promoter regions (*Chen et al., 2014*; *Dhar et al., 2016*; *Wang et al., 2010*), especially in the context of cancer (*Soto-Feliciano et al., 2023*). To determine the genomic binding patterns of MLL3 in HCC, we performed MLL3 chromatin immunoprecipitation (ChIP)-sequencing (ChIP-Seq) analysis in *Myc*; sg*Kmt2c* (sg*Kmt2c*.1 which generates heterozygous or homozygous indels) and *Myc*; sg*Trp53* liver cancer cell lines. Compared to the sg*Trp53* cells, sg*Kmt2c* cells had a marked reduction in MLL3 chromatin binding at a subset of genomic loci (*Figure 2A*). Approximately 40% of the peaks that were selectively lost in *Kmt2c*-deficient cells occurred at promoter regions, whereas unchanged MLL3 peaks between the two genotypes were more likely to be within intergenic regions (*Figure 2B*, *Figure 2—figure supplement 1*). Therefore, our data suggest that, beyond the canonical action of MLL3 at gene enhancers, MLL3 can also occupy promoter regions in *Myc*-induced liver cancer. Of note, the residual ChIP-seq signal observed in the sg*Kmt2c* cells most likely reflects the binding of MLL4 and/or remnant MLL3 since the antibody used in these experiments can recognize both MLL3 and MLL4 proteins (*Dorighi et al., 2017*). Nonetheless, the downregulated peak signals in *Myc*; sg*Kmt2c* cells were specifically due to MLL3 disruption.

Similar to the *Drosophila* Trithorax-related complex (*Schuettengruber et al., 2017*), the mammalian MLL3 and MLL4 complexes facilitate gene transcription by establishing permissive modifications

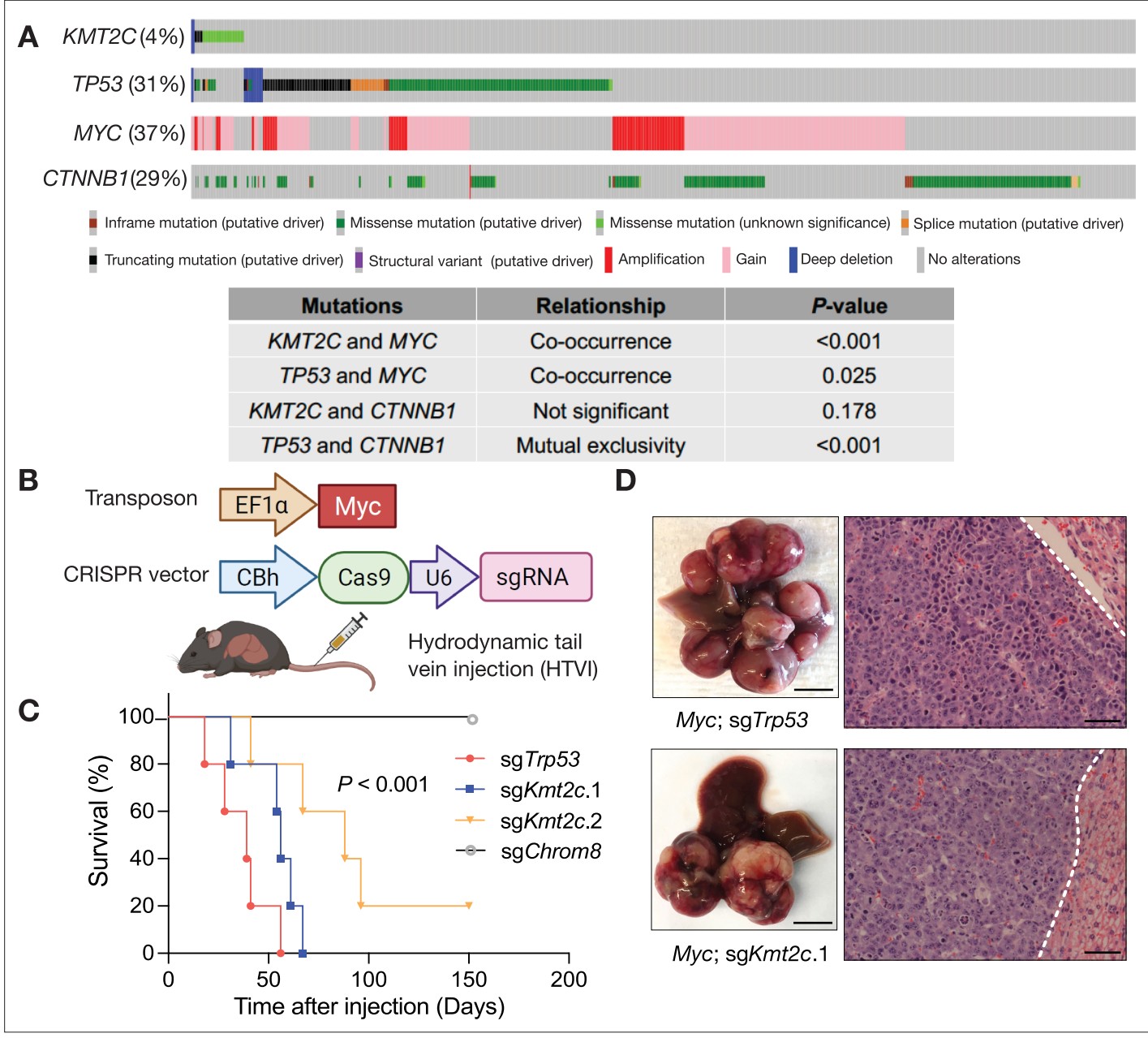

**Figure 1.** MLL3 constrains *Myc*-driven liver tumorigenesis. (**A**) Oncoprints displaying genomic mutations and deletions of *KMT2C* and *TP53*, gains and amplifications of *MYC*, and activating *CTNNB1* mutations in merged publicly available datasets (TCGA, MSK, INSERM, RIKEN, AMC, and MERCi) of 1280 sequenced hepatocellular carcinomas, and the table showing their relationships. p-Values were calculated by Fisher exact tests. (**B**) Schematic for hydrodynamic tail vein injection (HTVI) of gene delivery into murine livers. Vectors permitting stable expression of *Myc* transposon (top) and transient expression of Cas9 and single guide RNAs (sgRNAs) targeting putative tumor suppressors (bottom) via sleeping beauty transposase were introduced into hepatocytes by HTVI. (**C**) Survival curves of mice injected with *Myc* transposon and pX330 expressing two independent sgRNAs targeting *Kmt2c* after HTVI (*Myc*; sg*Kmt2c*.1, n=5; *Myc*; sg*Kmt2c*.2, n=5). *Myc*; sg*Trp53* (n=5), and *Myc*; sg*Chrom8* (n=5) serve as controls. Survival curves were compared using log-rank tests. (**D**) Representative images (left, liver macro-dissection, scale bar: 0.5 cm; right, H&E staining, scale bar: 100 μm) of mouse liver tumors generated by HTVI delivery of *Myc* transposon and in vivo gene editing. The dashed lines indicate the boundaries between liver tumors and non-tumor liver tissues.

The online version of this article includes the following source data and figure supplement(s) for figure 1:

**Figure supplement 1.** In vivo screen identifies MLL3 as a tumor suppressor in Myc-driven liver cancer.

**Figure supplement 2.** Suppression of *Kmt2c* by CRISPR or RNAi promotes *Myc*-driven liver cancer.

**Figure supplement 2—source data 1.** Original gel for surveyor assays in *Figure 1—figure supplement 2B*.

**Figure supplement 3.** MLL3 loss does not cooperate with CTNNB1 oncogene to drive liver cancer.

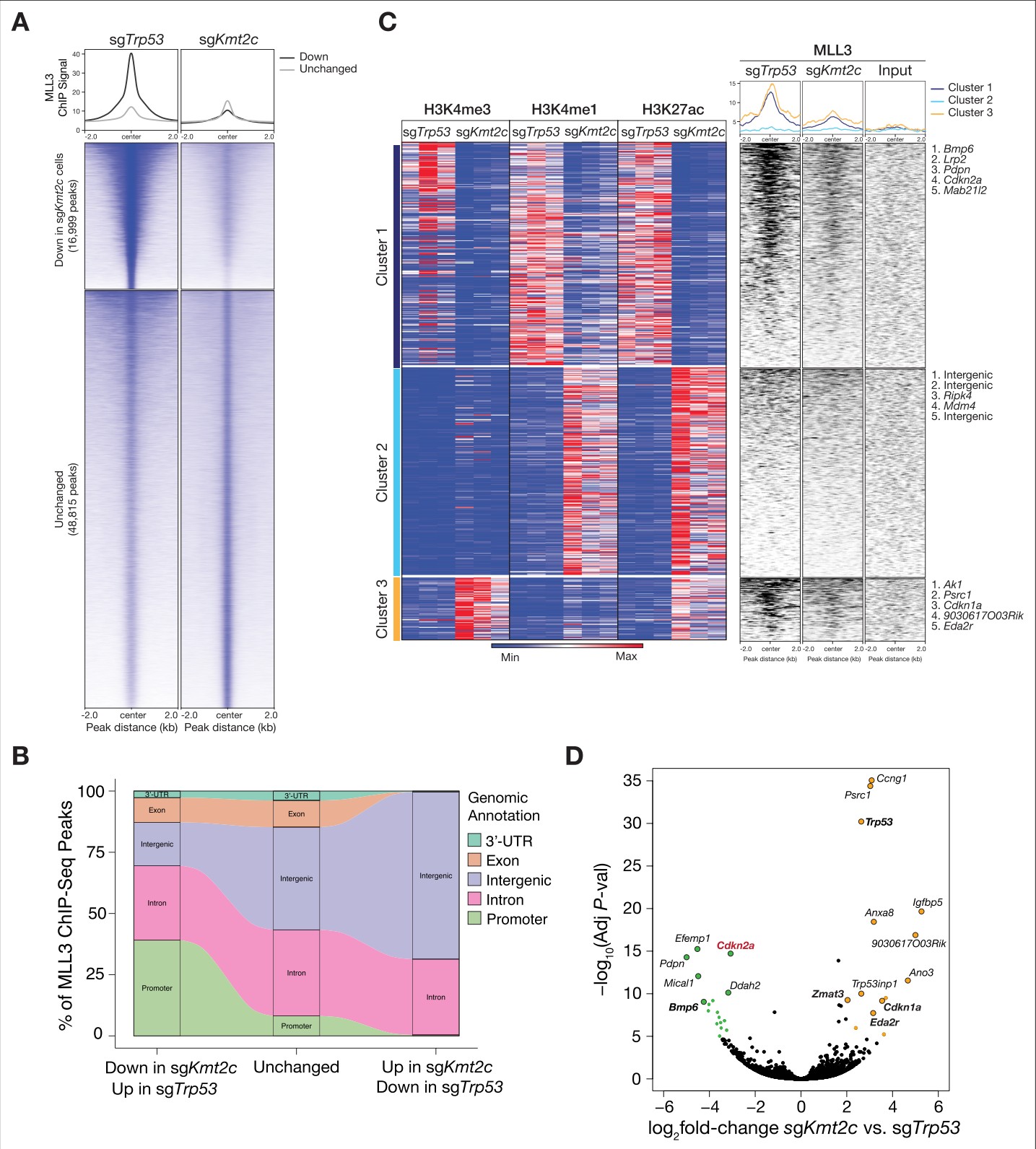

**Figure 2.** MLL3 disruption alters the chromatin and transcriptional landscape of liver cancer cells. (**A**) Tornado plots showing MLL3 chromatin immunoprecipitation-sequencing (ChIP-Seq) signal (peaks) that were down or remained unchanged in *Myc; sgKmt2c* cells relative to *Myc; sgTrp53* cells. Center: transcriptional start site (TSS). (**B**) Alluvial plot showing the percentages of MLL3 ChIP-Seq peaks in different genomic elements in *Myc; sgKmt2c* vs *Myc; sgTrp53* cells, including 16,999 peaks down, 48,815 peaks unchanged, and 265 peaks up in sg*Kmt2c* cells. Promoter regions were defined as

*Figure 2 continued on next page*

*Figure 2 continued*

TSS±2 kb. (**C**) Heatmaps of histone modification ChIP-Seq signals (H3K4me3, H3K4me1, H3K27ac, left panel) and MLL3 ChIP-Seq signal (right panel) at promoter or intergenic regions in three independent *Myc;* sg*Kmt2c* and *Myc;* sg*Trp53* liver tumor-derived cell lines. Cluster 1: loss of promoter and enhancer activity (loss of H3K4me3, H3K4me1, and H3K27ac); cluster 2: gain of enhancer activity (gain of H3K4me1 and H3K27ac); and cluster 3: gain of promoter activity (increase of H3K4me3). Representative top five loci for each cluster were listed on the right. (**D**) Volcano plot of differentially expressed genes revealed by RNA-sequencing of three independent *Myc;* sg*Kmt2c* and *Myc;* sg*Trp53* hepatocellular carcinoma (HCC) cell lines. Genes in sg*Kmt2c* cells with more than twofold expression change and exceeding adjusted p-value$<10^{-5}$ are color-labeled (orange: upregulated; green: downregulated). Some differentially expressed genes are labeled with gene symbols, and p53 targets are bolded.

The online version of this article includes the following figure supplement(s) for figure 2:

**Figure supplement 1.** MLL3 deficiency disrupts its binding at promoters in liver cancer cells.

**Figure supplement 2.** MLL3 disruption impacts transcriptional and histone modification profiles in liver tumors.

on histone H3K4 via the MLL3 and MLL4 methyltransferase (*Shilatifard, 2012*). To determine whether MLL3 disruption impacts the local or global chromatin landscape of HCC cells, we performed ChIP-Seq analyses for H3K4 methylation and H3K27 acetylation in six independently derived tumor cell lines: three each for *Myc;* sg*Kmt2c* and *Myc;* sg*Trp53* (*Figure 2C*). Cluster analysis on genomic areas revealed three clusters of genomic loci that showed enrichment or depletion between *Myc;* sg*Kmt2c* and *Myc;* sg*Trp53* tumor cells for each tested histone modification (*Figure 2C*, *Figure 2—figure supplement 2A–C*). Loci in cluster 1 (reduced H3K4me3, H3K4me1, and H3K27ac in sg*Kmt2c* cells) showed the most pronounced differences in chromatin modifications between the two liver tumor genotypes. In contrast, the loci in cluster 2 showed increased H3K4me1 and H3K27ac marks, most of which mapped to intergenic regions. The loci in cluster 3 showed increased H3K4me3 and included some p53 target genes such as *Cdkn1a* and *Eda2r*.

To determine whether these drastic changes in the chromatin landscape were associated with changes in MLL3 binding, we integrated the chromatin modifications results with our MLL3 ChIP-Seq results (*Figure 2C*). Interestingly, the loci in cluster 1, which displayed the most substantial changes in histone modifications, involved genes that showed enriched MLL3 binding in *Myc;* sg*Trp53* cells compared to the *Myc;* sg*Kmt2c* genotype. These data support a model whereby MLL3 binding to these loci facilitates the acquisition of a chromatin environment conducive for active gene transcription.

## MLL3 regulates specific tumor suppression programs in liver cancer cells

Transcriptional profiling helped hone in on potentially critical targets of MLL3. Specifically, we determined the output of these chromatin landscape changes by transcriptional profiling of the same set of *Myc;* sg*Trp53* and *Myc;* sg*Kmt2c* liver cancer cell lines described above. Despite the broad binding of MLL3 across the genome, we found only 248 differentially expressed genes (DEGs): 132 significantly upregulated (p<0.05, log$_2$ fold-change >2) and 116 significantly downregulated (p<0.05, log$_2$ fold-change <−2) in *Myc;* sg*Kmt2c* liver tumor cells compared to *Myc;* sg*Trp53* controls.

As predicted, transcripts encoding p53 and p53 target genes such as *Ccng1*, *Cdkn1a*, and *Zmat3* (*Bieging-Rolett et al., 2020*) were upregulated in *Myc;* sg*Kmt2c* cells, consistent with nonsense-mediated decay of truncated p53 transcripts and a concomitant reduction in p53 effector genes. Strikingly, some of the downregulated genes in *Myc;* sg*Kmt2c* lines mapped to loci enriched in cluster 1, including *Cdkn2a*, *Bmp6*, and *Lrp2* (*Figure 2C–D*, *Figure 2—figure supplement 2D, E*). Of note, sg*Kmt2c* did not lead to compensatory changes in the transcript levels of other major components of the COMPASS-like complexes, including Mll4 (*Kmt2d*), Utx (*Kdm6a*), Mll1 (*Kmt2a*), and Mll2 (*Kmt2b*; *Figure 3—figure supplement 1A*), suggesting that the alterations in MLL3 binding, histone modification, and transcription were specifically attributed to MLL3 disruption.

We reason that the mediators of MLL3 actions in tumor suppression should be within cluster 1 with reduced transcription and MLL3 binding in *Kmt2c*-deficient cells. To further characterize the gene repertoire directly regulated by MLL3 genomic binding, we integrated the results of MLL3 ChIP-seq and RNA-seq. Specifically, we selected downregulated DEGs that show concordant decreased binding of MLL3 in *Myc;* sg*Kmt2c* lines and subjected them to gene ontology analysis. Apart from the cluster 1 genes noted above, the integrative analysis revealed multiple MLL3-regulated tumor suppressive programs (*Figure 3A*, *Figure 3—figure supplement 1B*), including both cell-autonomous

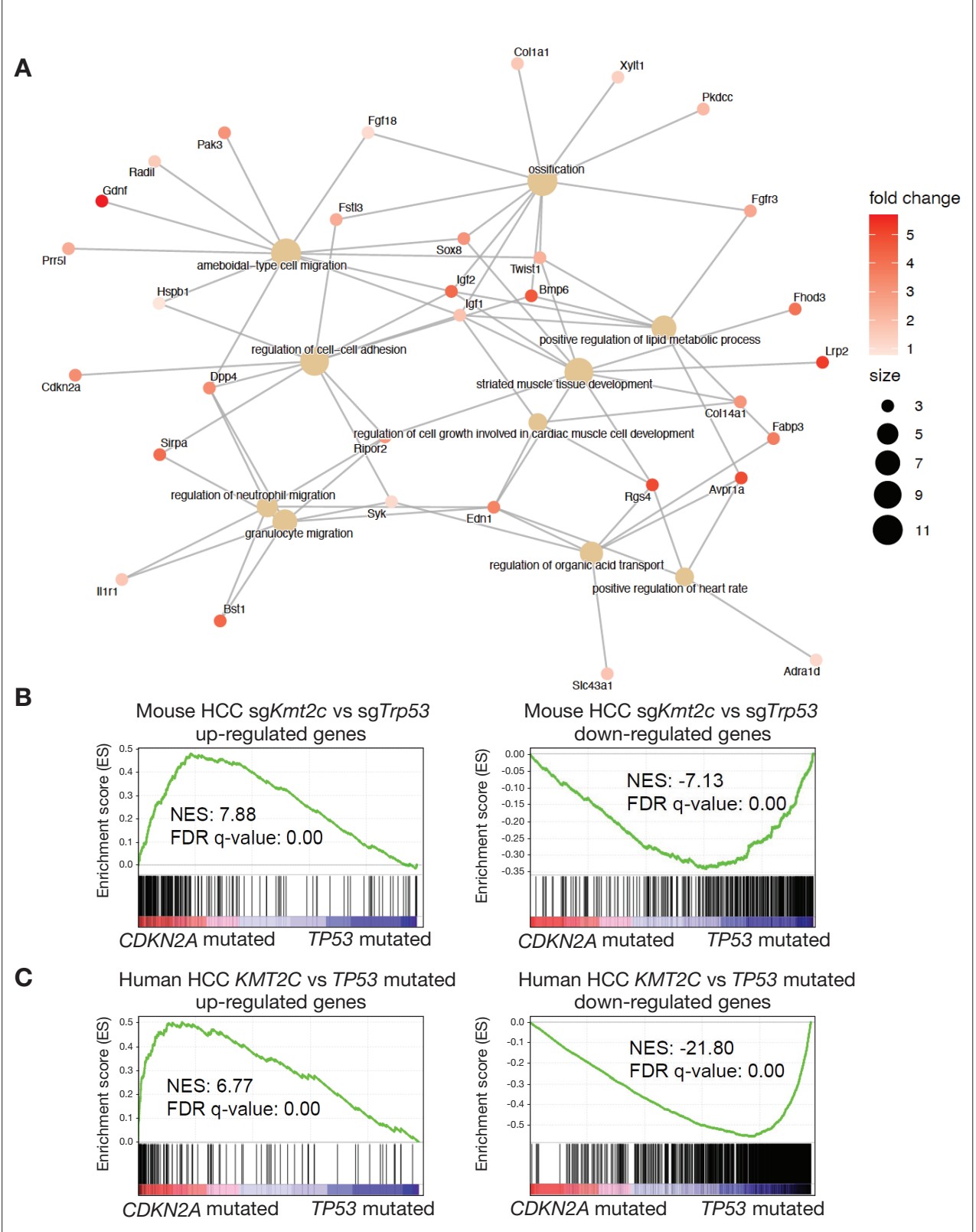

**Figure 3.** MLL3 regulates specific transcription programs including tumor suppressor CDKN2A. (**A**) Network plot showing the major biological processes and related genes directly regulated by MLL3 binding. p-Values and cluster sizes were calculated by the integrative analyses of RNA-seq and MLL3 chromatin immunoprecipitation-sequencing (ChIP-Seq), as detailed in the Materials and methods. (**B**) Gene set enrichment analysis (GSEA) plots of transcriptional signatures derived from mouse hepatocellular carcinomas (HCCs; *Myc*; sg*Kmt2c* vs *Myc*; sg*Trp53*) against transcriptomics of HCCs

*Figure 3 continued on next page*

*Figure 3 continued*

with *CDKN2A* vs *TP53* mutations. (**C**) GSEA plots of transcriptional signatures derived from *KMT2C* mutated/deleted human HCCs against the ones with *CDKN2A* mutations or homozygous deletions. HCCs with *TP53* mutations were used as the controls for both comparisons. Normalized enrichment scores (NES) and false discovery rate (FDR) q-values were calculated by GSEA.

The online version of this article includes the following figure supplement(s) for figure 3:

**Figure supplement 1.** MLL3 loss impacts specific transcription programs.

**Figure supplement 2.** *CDKN2A* and *KMT2C* mutations cause similar transcriptional changes in human hepatocellular carcinoma (HCC).

mechanisms (cellular metabolism) and non-autonomous mechanisms (interaction with extracellular matrix and immune system).

## *KMT2C* and *CDKN2A* mutations result in similar transcriptomes in human HCC

One genomic locus that stood out in our integrative analysis was *Cdkn2a*, which encompasses both the p16/Ink4a and p19/Arf (p14 in human) tumor suppressors (**Gil and Peters, 2006**). *CDKN2A* is located on the human chromosome 9p and is deleted or epigenetically silenced in many cancer types (**Sherr, 2012**), including HCC (2017). While MLL3 likely regulates a plethora of genes that contribute to its tumor-suppressive potential, the well-defined and potent antitumor functions of *Cdkn2a*-encoded proteins make them attractive candidates as functionally relevant MLL3 effectors.

Furthermore, in our analysis publicly available genomic data on 1280 HCC samples, we found that *CDKN2A* alterations, like *KMT2C* alterations, showed significant co-occurrence with *MYC* gains and amplifications. However, we were unable to conduct a meaningful test of mutual exclusivity between *CDKN2A* and *KMT2C* alterations (**Figure 3—figure supplement 2A**), given the constraints of sample size and the modest frequencies of alteration in each gene. Further dissection of transcriptional profiling datasets from human and mouse HCCs harboring known gene alterations using gene set enrichment analysis (GSEA) revealed that human tumors with *CDKN2A* deletions transcriptionally resembled both mouse and human HCC harboring *KMT2C* alterations (**Figure 3B and C**) but not those harboring *RB1* loss (**Figure 3—figure supplement 2B**), even though the tumor suppressor RB1 is regulated by CDKN2A/p16$^{INK4A}$ and their genomic alterations exhibit mutual exclusivity in multiple cancer types (**Knudsen et al., 2020**). While we cannot rule out the possibility that other factors drive these associations, our results support a biologically meaningful relationship between MLL3 and CDKN2A.

## *Cdkn2a* locus is a genomic and transcriptional target of MLL3 in liver cancer

To explore the relationship between MLL3 and *Cdkn2a* locus in more detail, we tested whether genes encoded by *Cdkn2a* were direct targets of MLL3-regulated transcription. Indeed, *Cdkn2a* is a cluster 1 locus that, in *Myc*; sg*Kmt2c* cancer cells, displays significant reduction in (1) expression, (2) H3K4me1/3 and H3K27ac levels, and (3) MLL3 binding at the *Cdkn2a* promoter compared with *Myc*; sg*Trp53* cells (**Figure 2C–D**, **Figure 4A**). Of note, MLL3 binding peaks were also observed within the gene body of *Cdkn2a*. The differential expression of Ink4a and Arf was confirmed by qPCR, immunoblotting, and ChIP-qPCR analyses on multiple *Myc*; sg*Kmt2c* and *Myc*; sg*Trp53* liver cancer lines (**Figure 4B**, **Figure 4—figure supplement 1A, B**). These results imply that *Cdkn2a* locus is a genomic and transcriptional target of MLL3 in liver cancer cells.

Since the *Myc*; sg*Trp53* and *Myc*; sg*Kmt2c* cells we studied above are not isogenic, we performed a series of additional experiments to demonstrate a direct transcriptional effect of MLL3 on the *Cdkn2a* locus. Because p53 inactivation can lead to compensatory increases in Ink4a and Arf expression (**Stott et al., 1998**), representing an alternative possibility accounting for the observed difference of *Cdkn2a* expression in sg*Trp53* vs sg*Kmt2c* cells. However, *p53* suppression in sg*Kmt2c* cells produced only a subtle and inconsistent effect on the expression of Ink4a and Arf, whereas *Kmt2c* suppression in sg*Trp53* cells consistently attenuated p16$^{Ink4a}$ and p19$^{Arf}$ protein levels (**Figure 4—figure supplement 1C, D**). As another means of ruling out the p53 pathway as an explanation for altered *Cdkn2a* expression in *Myc*; sg*Kmt2c* cells, we tested the ability of MLL3 to regulate *Cdkn2a* transcripts in an orthogonal liver cancer model driven by Myc and inactivation of Axin1, which is a well-defined

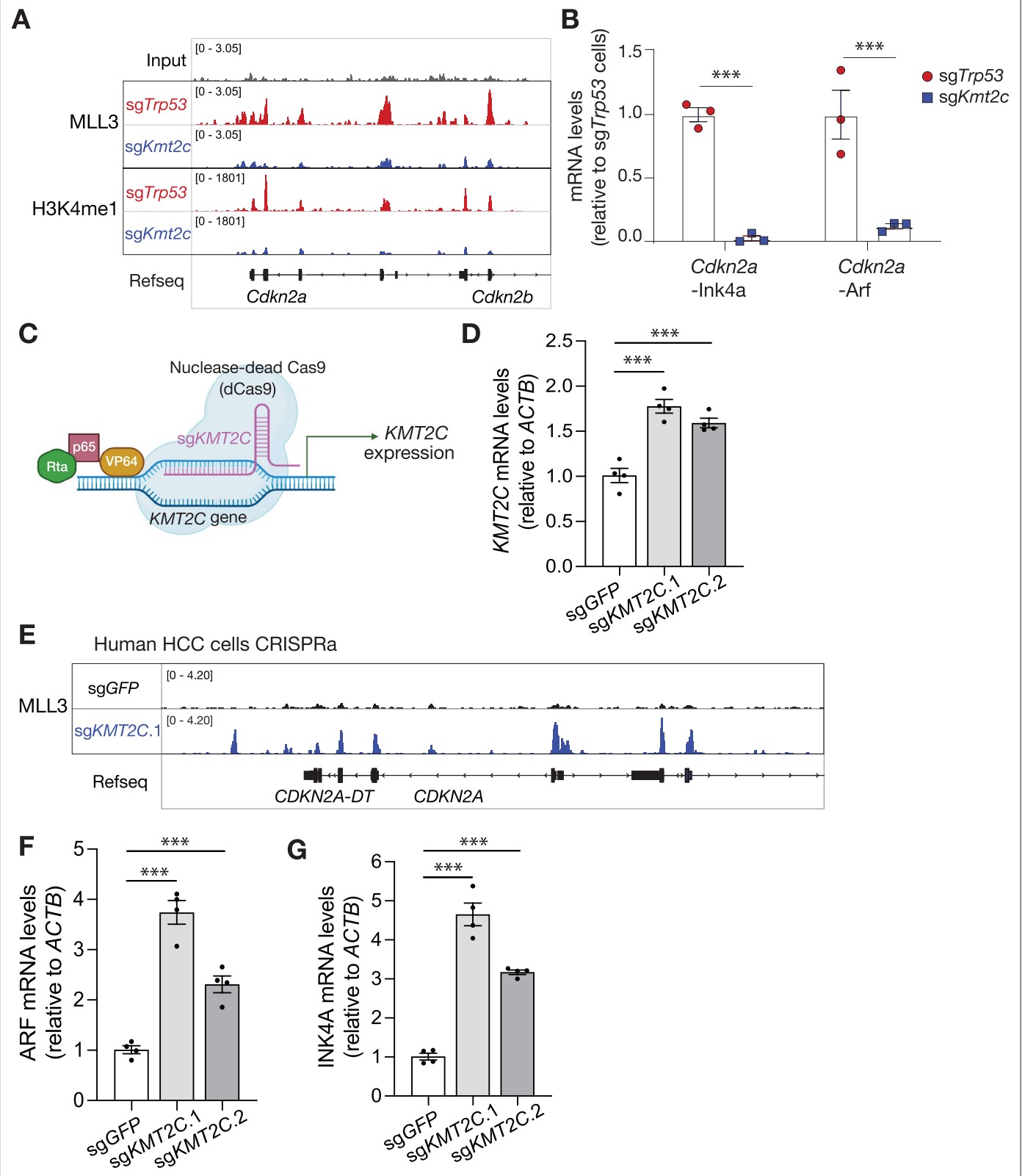

**Figure 4.** *CDKN2A* locus is a genomic and transcriptional target of MLL3 in liver cancer. (**A**) Genome browser tracks for MLL3 and H3K4me1 chromatin immunoprecipitation-sequencing (ChIP-Seq) in *Myc;* sg*Trp53* (red) and *Myc;* sg*Kmt2c* (blue) hepatocellular carcinoma (HCC) cell lines at the *Cdkn2a* locus. (**B**) qPCR analysis for mRNA expression of Ink4a and Arf from three independent *Myc;* sg*Kmt2c* and *Myc;* sg*Trp53* HCC lines (n=3 cell lines each genotype). Values are shown as mean ± SD. ***=p<0.001 (unpaired two-tailed t-test). (**C**) Schematic for CRISPR activation (CRISPRa) system of nuclease-

*Figure 4 continued on next page*

*Figure 4 continued*

dead Cas9 (dCas9) and VP64-p65-Rta (VPR) guided by sg*KMT2C* to activate *KMT2C* expression in human HLE HCC cell line. (**D**) qPCR analysis for mRNA expression of *KMT2C* in HLE cells with sg*GFP* (control) or two different CRISPRa single guide RNAs (sgRNAs) targeting *KMT2C* (n=4 cell lines each genotype). Each data point represents the average of technical duplicates. Data are shown as mean ± SEM. ***=p<0.001 (one-way ANOVA followed by post-hoc t-tests). (**E**) Genome browser tracks for MLL3 ChIP-Seq at the *CDKN2A* locus in HLE cells with sg*GFP* (control, black) or sg*Kmt2c*.1 (blue). (**F and G**) qPCR analysis for mRNA expression of (**F**) INK4A and (**G**) ARF in HLE cells with sg*GFP* (control) or sg*KMT2C* (n=4). Each data point represents the average of technical duplicates. Data are shown as mean ± SEM. ***=p<0.001 (one-way ANOVA followed by post-hoc t-tests).

The online version of this article includes the following source data and figure supplement(s) for figure 4:

**Figure supplement 1.** MLL3 directly regulates *Cdkn2a* expression in liver cancer cells.

**Figure supplement 1—source data 1.** Original western blots for *Figure 4—figure supplement 1A,C,D,E*.

tumor suppressor that negatively regulates β-catenin activity in HCC (*Satoh et al., 2000*). Liver cancer cells produced by hydrodynamic delivery of the *Myc* transposon vector and *Axin1* sgRNAs displayed reduced Ink4a and Arf expression upon *Kmt2c* knockdown without targeting p53 (*Figure 4—figure supplement 1E,F*). Importantly, MLL3 binding peaks at the *Cdkn2a* locus were also detected in *Myc*; sg*Axin1* liver cancer cells (*Figure 4—figure supplement 1G*), suggesting that *Cdkn2a* transcription is directly regulated by MLL3 rather than an indirect outcome of p53 loss. These data imply that MLL3 supports a chromatin environment at the *Cdkn2a* locus that facilitates the transcription of both Ink4a and Arf and raises the possibility that these factors contribute to the tumor suppressor activity of MLL3 in liver cancer.

We next set out to determine whether MLL3 binding is sufficient to induce transcriptional activation of the *CDKN2A* locus and, in doing so, extend our analysis to human liver cancer cells. As the *KMT2C* transcript is too large (14,733 bp) for cDNA transduction, we turned to the CRISPR activation (CRISPRa) system (*Chavez et al., 2015*) in a human hepatocellular carcinoma cell line (HLE). Following stable integration of the nuclease dead Cas9 fused to the VP64-p65-Rta (VPR) transcriptional activator, cells were transduced with two orthogonal sgRNAs targeting the human *KMT2C* promoter (or, as control, transduced with sgRNA against GFP; *Figure 4C*). Cells expressing the *KMT2C* sgRNAs showed a marked and specific increase in the expression of endogenous *KMT2C*, but not of *KMT2D* or *TP53* (*Figure 4D*, *Figure 5—figure supplement 1A, B*), which was accompanied by an increase in MLL3 binding to the *CDKN2A* locus (*Figure 4E*) and transcriptional upregulation of both *CDKN2A* transcripts (*Figure 4F and G*). Therefore, MLL3 directly binds and co-activates transcription of the *CDKN2A* locus in human liver cancer cells.

## MLL3 mediates oncogene-induced apoptosis in a *Cdkn2a*-dependent manner

The above results raise the possibility that the *Cdkn2a* products, INK4A and ARF, may contribute to the tumor suppressive activity of MLL3. In this regard, *Myc* overexpression in primary cells (mouse embryonic fibroblasts; MEFs) often triggers apoptosis (*Evan et al., 1992*), and this in turn limits tumorigenesis in a manner that is dependent on *Cdkn2a* (*Zindy et al., 1998*). This pathway also suppresses liver tumorigenesis since concomitant disruption of Ink4a and Arf using CRISPR, or germline deletion of Arf alone, cooperated with *Myc* overexpression to rapidly promote tumor development (*Figure 5—figure supplement 1C*). Similarly, *Kmt2c* suppression also attenuated MYC-induced apoptosis, as shown by tumor histology and apoptosis by TUNEL assay (*Negoescu et al., 1997*), 5 days after hydrodynamic delivery of transposon vectors encoding *Myc* together with GFP-linked shRNAs targeting *Kmt2c* (or *Renilla* luciferase as a control; *Figure 5A and B*). This difference in apoptosis correlated with an increase in retention of GFP-sh*Kmt2c* expressing cells 10 days after injection (*Figure 5—figure supplement 1D, E*). Altogether, these results show that *Kmt2c* suppression impairs *Myc*-induced apoptosis in vivo in a manner that is reminiscent of the anti-apoptotic effects of *Cdkn2a* loss in the context of aberrant *Myc* activation (*Eischen et al., 1999*; *Jacobs et al., 1999*; *Schmitt et al., 1999*).

To model the interaction between *Myc* overexpression, MLL3 function, and *Cdkn2a* regulation, we transduced liver progenitor cells (LPCs) with retroviral vectors encoding *Myc* linked to a reverse tetracycline transactivator (rtTA3), together with doxycycline (dox)-inducible *Kmt2c* shRNAs to enable reversible *Kmt2c* silencing (*Figure 5—figure supplement 2A*). Infection of LPCs with *Myc* in the

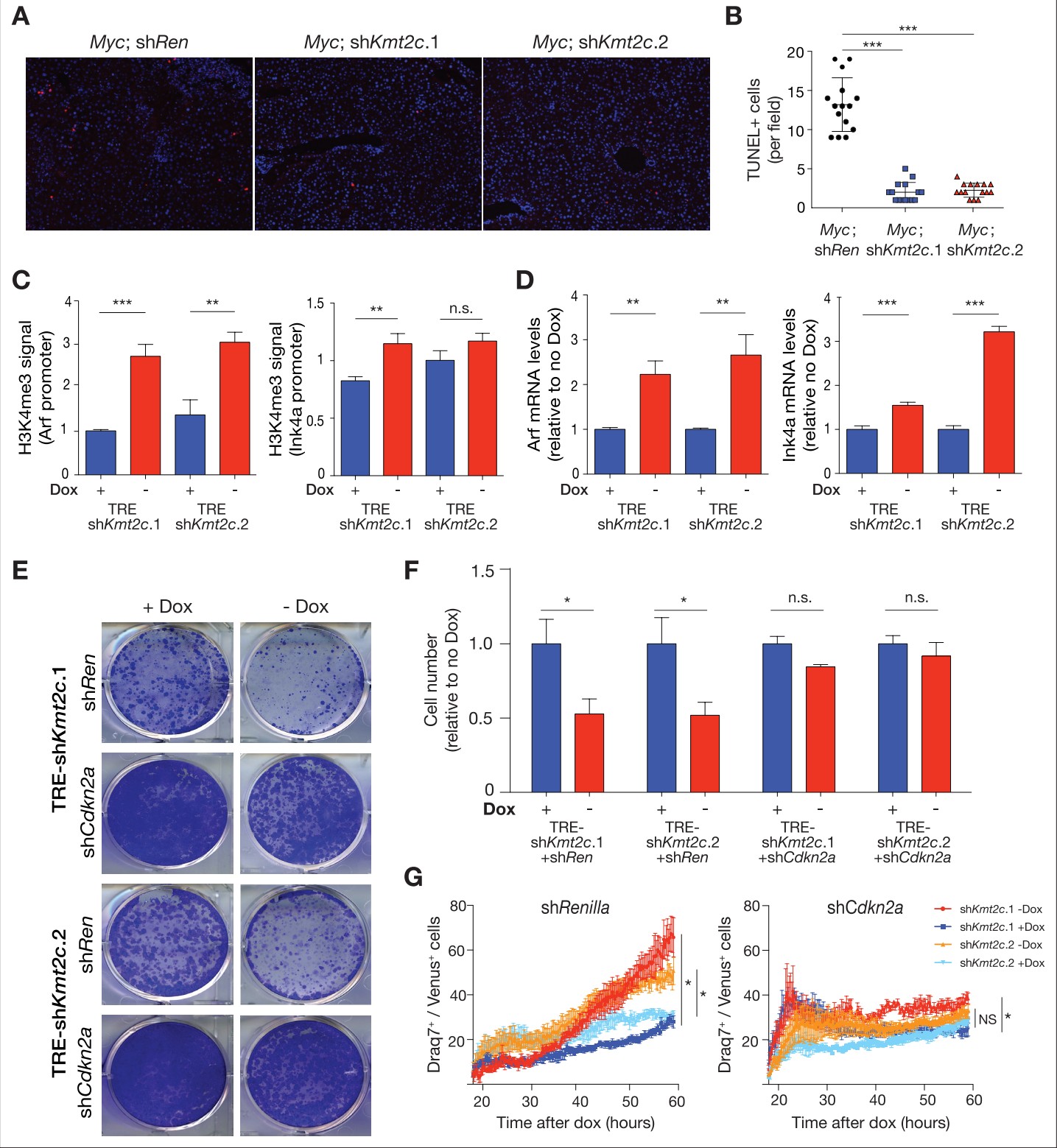

**Figure 5.** MLL3 mediates oncogene-induced apoptosis in a *Cdkn2a*-dependent manner. (**A**) Representative images of TUNEL-positive nuclei (red staining) in murine livers 5 days after hydrodynamic injection of the indicated vector combinations. DAPI(4′,6-diamidino-2-phenylindole) was used to visualize nuclei. (**B**) Quantification of TUNEL-positive nuclei in mouse livers 5 days after hydrodynamic tail vein injection (HTVI) of the indicated vector combinations. Data points represent the number of TUNEL-positive cells in five different high-power fields in three independent murine livers per group. ***=p<0.001 (one-way ANOVA followed by post hoc t-tests). (**C**) Chromatin immunoprecipitation (ChIP)-qPCR analysis for H3K4me3 signals at Arf

*Figure 5 continued on next page*

*Figure 5 continued*

and Ink4a promoters 4 days after doxycycline (dox) withdrawal in *Myc*-rtTA3; TRE-sh*Kmt2c* cells. Values are mean ± SD from technical replicates (n=3), and the experiments were conducted in two independent liver progenitor cell (LPC) lines with different sh*Kmt2c*. (**D**) qPCR analysis for mRNA expression of Arf and Ink4a 4 days after dox withdrawal in two independent LPC lines with different sh*Kmt2c*. Values are mean ± SD from technical replicates (n=3). \*\*\*=p<0.001 and \*\*=p<0.01 (unpaired two-tailed t-test). (**E**) Representative images of colony formation assay of the indicated cell lines 5 days after dox withdrawal. (**F**) Quantification of colony formation assay. Values are mean ± SD of three independent experiments with two independent LPC lines. \*=p<0.05 (unpaired two-tailed t-test). (**G**) Time course analysis of Draq7-positive (dead or permeabilized) cells as a fraction of Venus-positive, *Myc*-rtTA3; TRE-sh*Kmt2c* cells expressing constitutive shRNAs targeting both Ink4a and Arf (sh*Cdkn2a*) or *Renilla* luciferase (sh*Ren*) off and on dox. Values represent mean ± SEM of triplicate wells of each genotype at each timepoint of two independently derived LPC lines, infected with either sh*Ren* or sh*Cdkn2a*. \*=p<0.05 (unpaired two-tailed t-test of final average percentage Draq7$^+$/GFP$^+$). NS, not significant (p>0.05).

The online version of this article includes the following source data and figure supplement(s) for figure 5:

**Figure supplement 1.** *Kmt2c* suppression reduces cell clearance upon enforced *Myc* expression in vivo.

**Figure supplement 2.** Endogenous *Kmt2c* restoration triggers apoptosis and is accompanied by increased *Cdkn2a* expression.

**Figure supplement 2—source data 1.** Original western blots for *Figure 5—figure supplement 2B,F,G*.

presence of MLL3 (i.e. cells infected with *Myc*-rtTA3 and a dox-inducible shRNA targeting *Renilla* luciferase) acutely activated INK4A and ARF expression (*Figure 5—figure supplement 2B*), and these cells could not be maintained in culture. Phenocopying the ability of *Myc* and *Kmt2c* suppression to transform liver cells in vivo, combined *Myc* and sh*Kmt2c* expression facilitated the persistent growth of cells maintained on Dox (*Figure 5—figure supplement 2C,D*). By contrast, dox withdrawal induced *Kmt2c* mRNA expression and H3K4me3 deposition at the *Cdkn2a* promoters, ultimately leading to elevations in Arf and Ink4a mRNA and protein (*Figure 5C and D*), reduced colony formation, and increased apoptosis (*Figure 5—figure supplement 2D–F*). Furthermore, constitutive shRNA-mediated knockdown of Arf and Ink4a through targeting of the shared exon 2 (sh*Cdkn2a*) significantly rescued colony-forming capacity and prevented cell death following *Kmt2c* restoration as determined by time-lapse microscopy of cells cultured with a fluorescent dye that stains dead cells (*Figure 5E–G*, *Figure 5—figure supplement 2G*). These data support a model whereby a prominent tumor suppressive output of MLL3 in liver cancer involves direct upregulation of *Cdkn2a* that, when impaired, attenuates the MYC-induced apoptotic program and permits tumor progression.

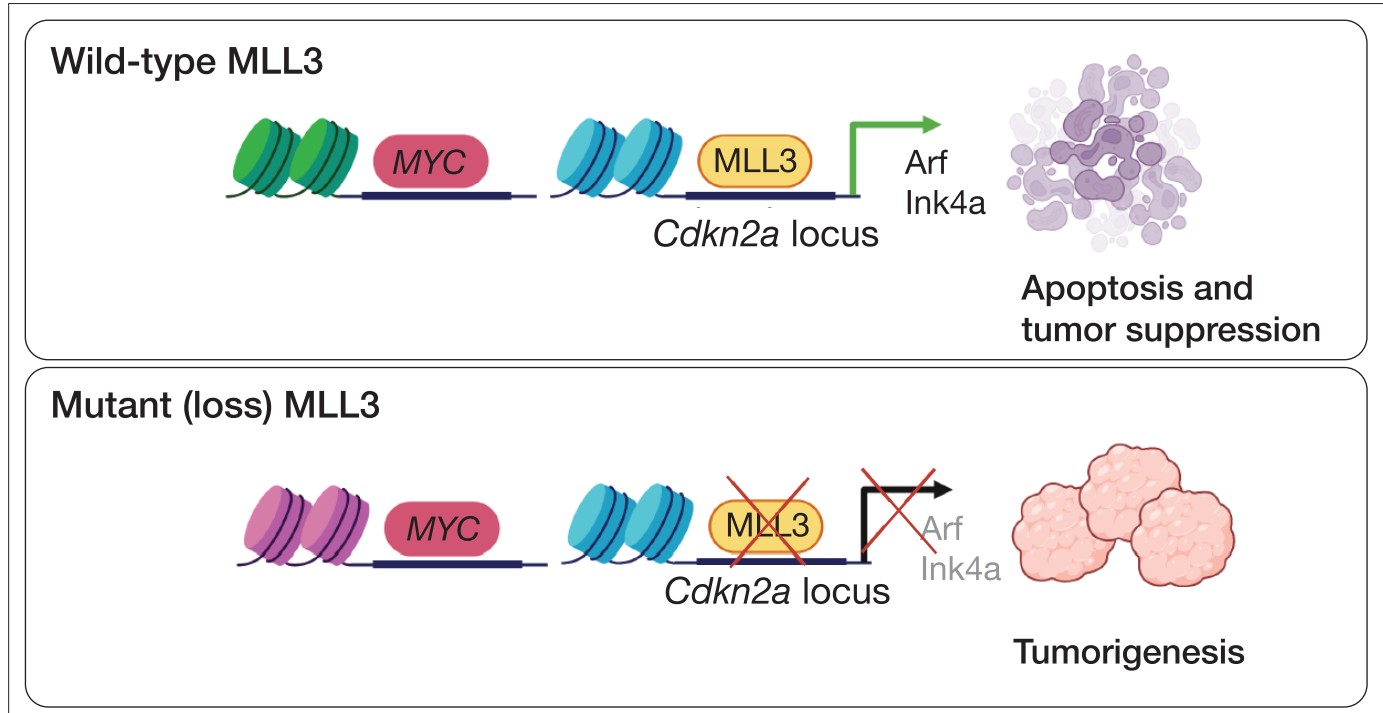

**Figure 6.** Model of MLL3 as a tumor suppressor in liver cancer. MLL3 restricts MYC-induced liver tumorigenesis by directly activating the *Cdkn2a* locus to mediate tumor cell apoptosis.

## Discussion

Our study combined genetic, epigenomic, and animal modeling approaches to identify *Cdkn2a* as an important regulatory target of MLL3 in both mouse and human liver cancers. Our results support a model whereby oncogenic stress, herein produced by MYC, leads to an increase in the binding of MLL3 to the *CDKN2A* locus, an event that is associated with the accumulation of histone marks linked to the biochemical activity of MLL3-containing complexes and conducive to gene activation (*Figure 6*). Accordingly, these events are accompanied by transcriptional upregulation of two key *Cdkn2a* gene products, Ink4a and Arf. Moreover, suppression of *Kmt2c* phenocopies the effects of *Cdkn2a* inactivation in abrogating MYC-induced apoptosis. Conversely, suppression of *Cdkn2a* diminishes the anti-proliferative effects of *Kmt2c* restoration. As such, our results establish a conserved epistatic relationship between the chromatin modifier MLL3 and a well-characterized tumor suppressor network.

The epistatic relationship described above might be expected to lead to mutual exclusivity of *KMT2C* and *CDKN2A* alterations; however, we did not observe significant mutual exclusivity in liver cancer, which is likely due to insufficient samples sizes needed to obtain statistical power. Alternatively, other functionally important components linked to the *CDKN2A* locus could produce *CDKN2A*-independent forces that drive selection for chromosome 9p deletions, including type I interferon genes, *CDKN2B*, and *MTAP* (*Barriga et al., 2022*). Alternatively, mutual exclusivity between *KMT2C* and *CDKN2A* alterations would be expected only under circumstances where *CDKN2A* action is the most dominant MLL3 effector. Indeed, it seems likely that multiple downstream genes, including factors involved in interactions with stromal and immune populations, contribute to MLL3-driven tumor suppression, and their relative importance may vary between cell and tissue types. Such a variable output in cancer-relevant gene regulation has been noted for other chromatin regulators that, at the extreme, serve as pro-oncogenic factors in some contexts and tumor suppressors in others (*Fountain et al., 1992*; *Schmid et al., 2000*; *Sun et al., 2017*; *Xia et al., 2021*). Furthermore, our observation that *Kmt2c* deficiency cooperated with MYC but not CTNNB1 to drive HCC highlights such context specificity and is in line with recent findings that chromatin context could favor particular oncogenic alterations over others (*Weiss et al., 2022*).

UTX (KDM6A), MLL3 (KMT2C), and MLL4 (KMT2D), the core catalytic components of the COMPASS-like complex, are all considered tumor suppressors, with frequent loss-of-function genomic alterations found in a broad spectrum of human cancers (*Revia et al., 2022*; *Sze and Shilatifard, 2016*). While each of these components regulates redundant sets of genes (*Hu et al., 2013*; *Lee et al., 2009*), they may exert their tumor suppressive functions through different mechanisms. In liver and pancreas cancer models, UTX can control the expression of negative regulators of mTOR such as *DEPTOR*, and its disruption prevents their transcription and facilitates tumorigenesis through increased mTORC1 activity (*Revia et al., 2022*). Additionally, while the mechanisms of MLL4 activity have not been examined in liver cancer, studies suggest that MLL4 suppresses skin carcinogenesis by promoting lineage stability and ferroptosis independently of MLL3 (*Egolf et al., 2021*). Our study demonstrates that MLL3 is both necessary and sufficient for efficient transcriptional activation of the *CDKN2A* locus that drives oncogene-induced apoptosis. The molecular basis for this heterogeneity in effector output remains to be determined, but it seems likely that different subsets of target genes are preferentially disabled by haploinsufficiency of individual components and/or subject to compensation by the remaining COMPASS complex activities. Systematic studies comparing the binding, histone modifications, and transcriptional output of cells across a spectrum of allelic configurations of COMPASS complex factors will be needed to achieve a more holistic understanding of their functions and interactions in different contexts.

The most well-established role for MLL3/4-UTX-containing complexes is the control of H3K4 monomethylation at enhancers during development (*Herz et al., 2010*; *Hu et al., 2013*). While our ChIP-Seq studies also revealed binding of MLL3/4 to enhancers in liver tumor cells, an even larger fraction of genes—including *Cdkn2a*—showed MLL3/4 chromatin enrichment at gene promoters, and indeed, transcription of this class of genes was most affected by *Kmt2c* disruption. Interestingly, *Kmt2c* suppression preferentially limited the MLL3/4 enrichment at promoters and shifted residual complex binding toward intergenic regions. Such dynamic regulation of distinct cis-acting elements by the MLL3/4 complex has also been observed in other contexts (*Cheng et al., 2014*; *Soto-Feliciano et al., 2023*), where the non-canonical binding of MLL3/4 at promoters is a recurrent tumor suppressive mechanism in cancer cells. MLL3/4 has also been observed to bind within the exons and introns, which

may enable chromatin looping of enhancers to activate gene expression (*Panigrahi and O'Malley, 2021*). Further studies into the action and regulation of MLL3/4 complexes at promoters and gene bodies will be informative and may yield new insights into the actions of the COMPASS-like complex in cancer.

While *CDKN2A* showed a surprisingly dominant role in mediating the tumor-suppressive effects of the broadly acting MLL3 enzyme, there are precedents for a predominant contribution of a single gene to the functional output of chromatin-complex disruption. Indeed, polycomb repressive complexes (PRCs) broadly repress gene expression in different cell types through the coordinated action of PRC1 and PRC2 complexes that deposit and maintain repressive H3K27me3 marks on the enhancers of target genes, including *CDKN2A* (*Bracken et al., 2007*; *Kotake et al., 2007*). Despite these similarly broad effects, *CDKN2A* is often the most functionally relevant target of PRC-mediated repression, as genetic deletion of either the PRC1 component *Bmi1* or the PRC2 component *Ezh2*, or treatment with small molecule inhibitors of EZH2, can facilitate *Cdkn2a* induction in normal and tumor cells. This, in turn, triggers anti-proliferative responses that can be rescued by *Cdkn2a* deletion (*Jacobs et al., 1999*; *Richly et al., 2011*). Notably, the COMPASS-like complexes are biochemically and functionally similar to Trithorax complexes in *Drosophila*, which have an evolutionarily conserved antagonistic relationship with PRC1 and PRC2 that controls epigenetic memory and cell fate during development (*Mills, 2010*; *Piunti and Shilatifard, 2016*). Our findings suggest such antagonism extends to tumor suppression in mammalian cells, likely via regulation of *Cdkn2a* and other tumor suppressor genes (*Soto-Feliciano et al., 2023*).

## Materials availability statement

Source files of all original gels and western blots were provided for the following figures:

> *Figure 1—figure supplement 2B*;
> *Figure 4—figure supplement 1A, C, D, E*;
> *Figure 5—figure supplement 2B, F, G*.

RNA sequencing and ChIP-Seq data files that support the findings of this study have been deposited in the Gene Expression Omnibus under the accession code GSE85055, as well as in the Dryad digital repository (doi:10.5061/dryad.7pvmcvdwm; doi:10.5061/dryad.f1vhhmh0h). Sequences of sgRNAs, shRNAs, and primers used in this manuscript are included in the *Supplementary file 1*.

## Materials and methods

### Animal experiments

8- to 10-week-old female C57BL/6 animals were purchased from Envigo (formerly Harlan). Each experiment was performed in mice from the same order. Arf-null animals (C57BL/6 background), originally provided by Dr. Charles Sherr, St. Jude Children's Research Hospital, were maintained in our breeding colony. For HTVI, a sterile 0.9% NaCl solution/plasmid mix was prepared containing oncogene transposons (5 µg DNA of pT3-*Myc* or 10 µg pT3-*Ctnnb1 N90*) with either 20 µg of pX330 expressing the indicated sgRNAs or 20 µg of pT3-EF1a-GFP-miRE plasmid together with CMV-SB13 Transposase (1:5 ratio). Mice were randomly assigned to experimental groups and injected with the 0.9% NaCl solution/plasmid mix into the lateral tail vein with a total volume corresponding to 10% of body weight in 5–7 s as described before (*Largaespada, 2009*; *Moon et al., 2019*; *Tschaharganeh et al., 2014*; *Xue et al., 2014*). Injected mice were monitored for tumor formation by abdominal palpation. All animal experiments were approved by the Memorial Sloan Kettering Cancer Center (MSK) Institutional Animal Care and Use Committee (protocol 11-06-011). Animals were monitored for signs of ill health by veterinary staff at the Research Animal Resource Center at MSK, and efforts were made to minimize suffering.

### Vector constructs

The pT3-*Myc* vector Addgene (#92046) and pT3-EF1a-GFP-miRE plasmid were described before (*Huang et al., 2014*). The pT3-*Ctnnb1 N90* vector (*Tward et al., 2007*) was obtained from Addgene (#31785). For CRISPR/Cas9-mediated genome editing, sgRNAs were subcloned into pX330 (Addgene, #42230; *Hsu et al., 2013*). All shRNA and sgRNA sequences are listed in *Supplementary file 1*.

## Derivation of primary liver tumor cell lines

Liver tumors were resected with sterile instruments, and 10–50 mg of tumor tissue was minced and washed in sterile PBS, incubated in a mix of 1 mg/mL collagenase IV and 3 mg/mL dispase (dissolved in sterile, serum-free DMEM(Dulbecco's Modified Eagle Medium)) with gentle shaking, washed with PBS, incubated for 5 min in 0.05% (w/v) trypsin, and washed and plated in complete DMEM (10% FBS(fetal bovine serum), 1× penicillin/streptomycin) on collagen-coated plates (PurCol, Advanced Biomatrix). Primary cultures were passaged until visibly free from fibroblasts. Cell lines were authenticated on a routine basis using short tandem repeat profiling, as well as tested for mycoplasma contamination and immediately discarded upon a positive test.

## Analysis of CRISPR-directed mutations

CRISPR-mediated insertions and deletions were detected by surveyor assay as directed by the manufacturer (Transgenomic/IDT). Briefly, after overnight lysis of primary tumors and cell lines at 37°C in buffer containing 0.4 mg/mL proteinase K, 10 mM Tris, 100 mM NaCl, 10 mM EDTA, and 0.5% SDS, pH 8.0, genomic DNA was extracted by isopropanol precipitation. ~250–500 bp regions flanking predicted CRISPR cleavage sites were PCR amplified with Herculase II taq polymerase, column purified (Qiagen), heated to 95°C, and slowly cooled to promote annealing of heteroduplexes. Following treatment with Surveyor nuclease, products were analyzed by electrophoresis on a 2% polyacrylamide gel. Primers used for surveyor assay are listed in *Supplementary file 1*. Amplified PCR products were separately gel purified and ligated into blunt-end digested pBlueScript (Stratagene). DNA from 48 transformed colonies was analyzed by Sanger sequencing using a T7 primer.

## CRISPR activation

Human HCC cell line HLE, purchased from JCRB Cell Bank (JCRB0404), was transduced by the lentivirus expressing nuclease-dead Cas9 (dCas9) fused with VPR (*Chavez et al., 2015*) and sgRNAs against *KMT2C* (sequence in *Supplementary file 1*) to generate stable MLL3 CRISPRa HLE line by puromycin selection.

## Generation and modification of primary cells

LPCs from E13.5–15.5 C57BL/6 embryos were isolated and grown in hepatocyte growth media (HGM) as previously described (*Zender et al., 2005*). To simultaneously overexpress *Myc* and conditionally suppress *Kmt2c*, LPCs were co-infected with a retroviral construct constitutively expressing both *Myc* and a reverse tet-transactivator (rtTA) (MSCV-*Myc*-IRES-rtTA) along with retroviral TRMPV vectors (MSCV-TRE-dsRed-miR30/shRNA-PGK-Venus-IRES-NeoR) (*Zuber et al., 2011*) expressing ds-Red linked, teint-responsive shRNAs targeting *Kmt2c* cloned into an optimized mir-30 context ('mir-E,' TRPMVe; *Zuber et al., 2011*). For selection of infected cells and sustained sh*Kmt2c* expression, cells were maintained in HGM with neomycin (1 mg/mL) and dox (1 µg/mL) starting 2 days after infection. To introduce constitutively expressed shRNAs in the setting of inducible sh*Kmt2c*, retroviral MLPe vectors (MSCV-LTR-mir-E-PGK-Puro-IRES-GFP; *Dickins et al., 2005*). GFP-linked shRNAs targeting either *Cdkn2a* or *Renilla* luciferase (as control) were co-infected with MSCV-*Myc*-IRES-rtTa and TRMPVe-sh*Kmt2c*. Triple-infected cells were maintained in media with neomycin, puromycin (2 µg/mL), and dox 2 days post infection. Infected continuously proliferating cells were transitioned to growth in complete DMEM and maintained on collagen-coated plates.

## Colony assays

For measurement of cell proliferation, 5000 transduced and selected LPCs or MEFs were plated in triplicate in 6-well plates. Tetracycline-inducible sh*Kmt2c*–expressing LPCs were grown in the presence or absence of dox, and after 5 days, cells were fixed with formalin and methanol and stained with 0.05% crystal violet. MEFs were fixed after 6 days with formalin and methanol and stained with 0.05% crystal violet.

## Apoptosis assays

Apoptosis was measured in LPCs via Annexin V staining according to the manufacturer's instructions (eBiosciences, Annexin-V APC). 25,000 cells were grown with and without dox for 3 days, trypsinized,

washed with Annexin-V binding buffer, and ~100,000 cells were incubated with Annexin-V APC and analyzed on an LSRII flow cytometer (BD).

## Live imaging

Imaging was performed on LPCs immortalized by linked overexpression of *Myc* and two independent, inducible *Kmt2c* shRNAs constitutively expressing shRNAs targeting *Renilla* luciferase or *Cdkn2a* (generated as detailed above). 1000 cells were plated on collagen-coated, 96 well, clear bottom imaging plates in media supplemented with 300 nM Draq7 (Invitrogen) with and without dox, in triplicate by genotype. 18 hr after plating cells, Venus (marking all plated cells) and Draq7 fluorescence was collected in two, 10× fields of each well every 15 min for 41 hr using an automated, high content microscope (InCell 6000, General Electric).

## Chromatin immunoprecipitation

Histone ChIP was performed as previously described (*Lee et al., 2006*). Briefly, cell samples were cross-linked in 1% formaldehyde for 10 min, and the reaction was stopped by addition of glycine to 125 mM final concentration. Fixed cells were lysed in SDS lysis buffer, and the chromatin was fragmented by sonication (Covaris). Sheared chromatin was incubated with antibodies (final concentration 10 µg/mL) against H3K4me3 (Abcam, ab8580, Lot:GR164706-1), H3K27ac (Abcam, ab4729, Lot:GR200563-1), or H3K4me1 (Abcam; ab8895, Lot:GR114265-2) or with normal rabbit IgG (Abcam, ab46540) at 4°C for overnight. Antibodies were recovered by binding to protein A/G agarose (Millipore), and the eluted DNA fragments were used directly for qPCR or subjected to high-throughput sequencing (ChIP-Seq) using a HiSeq 2000 platform (Illumina). High-throughput reads were aligned to mouse genome assembly NCBI37/mm9 as previously described (*Barradas et al., 2009*). Reads that aligned to multiple loci in the mouse genome were discarded. The ChIP-Seq signal for each gene was quantified as total number of reads per million in the region 2 kb upstream to 2 kb downstream of the transcription start site (TSS). Primers used for ChIP-qPCR of mouse *Cdkn2a* promoter (*Barradas et al., 2009*) are listed in Table S1.

The complete dataset is available at NCBI Gene Expression Omnibus (GSE85055), as well as the Dryad digital repository (doi:10.5061/dryad.7pvmcvdwm).

For the MLL3 ChIP-Seq, the following protocol was used. Cross-linking ChIP in mouse and human HCC cells was performed with 10–20×10⁷ cells per immunoprecipitation. Cells were collected, washed once with ice-cold PBS, and flash-frozen. Cells were resuspended in ice-cold PBS and cross-linked using 1% paraformaldehyde (PFA; Electron Microscopy Sciences) for 5 min at room temperature with gentle rotation. Unreacted PFA was quenched with glycine (final concentration 125 mM) for 5 min at room temperature with gentle rotation. Cells were washed once with ice-cold PBS and pelleted by centrifugation (800 *g* for 5 min). To obtain a soluble chromatin extract, cells were resuspended in 1 mL of LB1 (50 mM HEPES pH 7.5, 140 mM NaCl, 1 mM EDTA, 10% glycerol, 0.5% NP-40, 0.25% Triton X-100, and 1× complete protease inhibitor cocktail) and incubated at 4°C for 10 min while rotating. Samples were centrifuged (1400 *g* for 5 min), resuspended in 1 mL of LB2 (10 mM Tris-HCl pH 8.0, 200 mM NaCl, 1 mM EDTA, 0.5 mM EGTA, and 1× complete protease inhibitor cocktail), and incubated at 4°C for 10 min while rotating. Finally, samples were centrifuged (1400 *g* for 5 min) and resuspended in 1 mL of LB3 (10 mM Tris-HCl pH 8.0, 100 mM NaCl, 1 mM EDTA, 0.5 mM EGTA, 0.1% sodium deoxycholate, 0.5% N-lauroylsarcosine, and 1× complete protease inhibitor cocktail). Samples were homogenized by passing seven to eight times through a 28-gauge needle, then Triton X-100 was added to a final concentration of 1%. Chromatin extracts were sonicated for 14 min using a Covaris E220 focused ultrasonicator. Lysates were centrifuged at 20,000 *g* for 10 min at 4°C, and 5% of the supernatant was saved as input DNA. Beads were prepared by incubating them in 0.5% BSA in PBS and antibodies overnight (100 µL of Dynabeads Protein A or Protein G [Invitrogen] plus 20 µL of antibody). The antibody was anti-MLL3/4, which was kindly provided by the Wysocka laboratory (*Dorighi et al., 2017*). Antibody-beads mixes were washed with 0.5% BSA in PBS and then added to the lysates overnight while rotating at 4°C. Beads were then washed six times with RIPA buffer (50 mM HEPES pH 7.5, 500 mM LiCl, 1 mM EDTA, 0.7% sodium-deoxycholate, and 1% NP-40) and once with TE-NaCl Buffer (10 mM Tris-HCl pH 8.0, 50 mM NaCl, and 1 mM EDTA). Chromatin was eluted from beads in Elution buffer (50 mM Tris-HCl pH 8.0, 10 mM EDTA, and 1% SDS) by incubating at 65°C for 30 min while shaking, supernatant was removed by centrifugation, and crosslinking was reversed by

further incubating chromatin overnight at 65°C. The eluted chromatin was then treated with RNase A (10 mg/mL) for 1 hr at 37°C and with proteinase K (Roche) for 2 hr at 55°C. DNA was purified by using phenol-chloroform extraction followed with ethanol precipitation. The NEBNext Ultra II DNA Library Prep kit was used to prepare samples for sequencing on an Illumina NextSeq500 (75 bp read length, single-end, or 37 bp read length, and paired-end).

The complete dataset for MLL3 ChIP-Seq is available at the Dryad digital repository (doi:10.5061/dryad.f1vhhmh0h).

## Immunoblotting

Cell pellets were lysed in Laemmli buffer (100 mM Tris-HCl pH 6.8, 5% glycerol, 2% SDS, and 5% 2-mercaptoethanol). Equal amounts of protein were separated on 12% SDS–polyacrylamide gels and transferred to PVDF(polyvinylidene difluoride) membranes (90 V, 75 min). β-actin was used as a control to ensure equal loading, and images were analyzed using the AlphaView software (ProteinSimple). Immunoblotting was performed using antibodies for MYC (1:1000, Abcam, ab32072), p53 (1:500, Leica Biosystems, NCL-p53-505), p19 (1:250, Santa Cruz Biotechnology, sc-32748), p16 (1:250, Santa Cruz Biotechnology, sc-1207), Axin1 (1:1000, Cell Signaling, #2074), and β-actin (1:10000, Sigma-Aldrich, clone AC-15). Source files of all western blots were provided for *Figure 1—figure supplement 2*, *Figure 4—figure supplement 1*, *Figure 5—figure supplement 2*.

## Quantitative RT-PCR

Total RNA was isolated using RNeasy Mini Kit, QIAshredder Columns, and RNase-Free DNase Set (Qiagen). cDNA synthesis was performed using TaqMan Reverse Transcription Reagents (Thermo Fisher Scientific). Real-time PCR was carried out using Power SYBR Green Master Mix (Thermo Fisher Scientific) and the Life Technologies ViiA 7 machine. Transcript levels were normalized to the levels of mouse or human *Actb* mRNA expression and calculated using the ΔΔCt method. Each qRT-PCR was performed in triplicate using gene-specific primers (sequences listed in Table S1).

## RNA sequencing and differential expression analysis

For RNA sequencing, total RNA from three independent tumor-derived cell lines (*Myc*; sg*Trp53* and *Myc*; sg*Kmt2c*) was isolated using RNeasy Mini Kit, QIAshredder Columns and RNase-Free DNase Set (Qiagen). RNA-Seq library construction and sequencing were performed according to protocols used by the integrated genomics operation Core at MSK. 5–10 million reads were acquired per replicate sample. After removing adaptor sequences with Trimmomatic, RNA-seq reads were aligned to GRCm38.91(mm10) with STAR (*Dobin et al., 2013*). Genome-wide transcript counting was performed by HTSeq to generate an FPKM(Fragments Per Kilobase per Million mapped fragments) matrix (*Anders et al., 2015*). DEGs were identified by DESeq2 (v.1.8.2, package in R) and plotted in the volcano plot. The complete dataset is available at NCBI Gene Expression Omnibus (GSE85055).

## Integrative analyses of RNA-seq and MLL3 ChIP-seq

Differential peaks from ChIP-Seq data were annotated by assigning all intragenic peaks to that gene while intergenic peaks were assigned using linear genomic distance to the TSS. Genes that were coordinately regulated (fold change >1.5 and adjusted p-value<0.1) in MLL3 ChIP-seq and RNA-seq data were selected for the integrated analysis. Enriched pathways were scored using the enrichGO function with 'biological process' in the clusterProfiler R package. Redundant pathways were collapsed using the 'simplify' function with a cutoff of 0.7 with the p.adjust metric. Network analysis was performed using differential peaks and genes by running enrichplot::cnetplot in R with default parameters.

## Human cancer analyses

RNA sequencing data of selected samples with somatic mutations or homozygous deletions of *KMT2C*, *CDKN2A*, *TP53*, or *RB1* in the TCGA HCC dataset were downloaded from Broad Institute TCGA Genome Data Analysis Center. To obtain transcriptional signatures of HCC with genomic mutations and deletions of either *KMT2C*, *CDKN2A*, and *RB1*, differential gene expression analyses were performed by DESeq2 (with *TP53*-mutated HCCs as controls). The oncoprints of homozygous deletions and somatic mutations of *KMT2C*, *CDKN2A*, and *TP53*, as well as MYC gains and amplifications from human HCC datasets (*Cancer Genome Atlas Research Network, 2017*, MSK [*Harding et al.,*

*2019*; *Zheng et al., 2018*], INSERM [*Schulze et al., 2015*], RIKEN [*Fujimoto et al., 2012*], AMC [*Ahn et al., 2014*], and MERCi [*Ng et al., 2022*]) were generated by cBioPortal (https://www.cbioportal.org ; *Cerami et al., 2012*; *Gao et al., 2013*).

## Gene set enrichment analysis

GSEA was performed using the GSEAPreranked tool for conducting GSEA of data derived from RNA-seq experiments (version 2.07) against other signatures. The metric scores (normalized enrichment scores and false discovery rate q-values) were calculated using the sign of the fold change multiplied by the inverse of the p-value (*Subramanian et al., 2005*). Specifically, transcriptional signatures were derived based on significantly changed genes (p-adjusted<0.05, absolute fold change >2) from RNA-seq of mouse HCC cell lines (*Myc*; sg*Kmt2c* vs *Myc*; sg*Trp53*, n=3 each genotype), and in human HCCs with mutations in *KMT2C* vs *TP53* (p-adjusted<0.05, absolute fold change >2). These signatures were compared to the transcriptional comparison of TCGA human HCCs with genomic inactivation of *CDKN2A* vs *TP53*.

## Statistical analyses

Data are presented as mean ± D or SEM as specified. The statistical comparison between two groups was accomplished with the two-tailed student's t-test or one-way ANOVA followed by post hoc t-tests among three or more groups. The analyses for co-occurrence or mutual exclusivity of mutations were performed using Fisher Exact test. Comparisons of survival curves were performed by log-rank tests. All statistical tests were performed using the Prism 8 software. All data presented in the manuscript have been replicated in independent cohorts of mice or in at least three biological replicates for in vitro experiments. On the basis of predicted effects of oncogene-tumor suppressor interaction introduced by HTVI in mice, with a power of 0.8 and p<0.05, we calculated a minimum sample size of 5 mice per group. Animals within the same cage were randomly allocated into control and experimental groups, with the group assignment recorded in a master spreadsheet and unmasked only when all samples of the respective experiments were analyzed. Data collection of each experiment was detailed in the respective figures, figure legends, and methods. No data were excluded from studies in this manuscript.

## Acknowledgements

We thank Charles Sherr and Janet Novak for constructive guidance and advice on all aspects of this study. We thank Ali Shilatifard, Lu Wang, and all members of the Lowe lab for helpful and stimulating discussions. We gratefully thank A Chramiec for excellent technical assistance. We thank Joanna Wysocka (Stanford University) for kindly sharing the anti-MLL3/4 antibody used in our ChIP-Seq experiments. This work was supported by grants to SWL (P01 CA013106 and R01 CA233944) from the NIH/NCI, as well as by the National Center for Tumor Disease, Heidelberg, and grants of the German Research Foundation to DFG (SFB/TRR77). This work was also supported by the NIH/NCI Cancer Center Support Grant to Memorial Sloan Kettering Cancer Center (P30 CA008748). YMSF is supported by a MOSAIC K99/R00 Award from the NIH/NIGMS (1K99GM140265-01). CZ is supported by an F32 Postdoctoral Fellowship (1F32CA257103) from the NIH/NCI. JPM was a recipient of a Postdoctoral Fellowship (PF-14-066-01-TBE) from the American Cancer Society. DFT is supported by a Young Investigator Group (VH-NG-1114) by the Helmholtz foundation. SWL is the Geoffrey Beene Chair for Cancer Biology and an investigator of the Howard Hughes Medical Institute.

## Additional information

### Competing interests

C David Allis: is a co founder of Chroma Therapeutics and Constellation Pharmaceuticals and a Scientific Advisory Board member of EpiCypher. Scott W Lowe: is an advisor for and has equity in the following biotechnology companies: ORIC Pharmaceuticals, Faeth Therapeutics, Blueprint Medicines, Geras Bio, Mirimus Inc, Senescea, and PMV Pharmaceuticals. S.W.L. also acknowledges receiving funding and research support from Agilent Technologies and Calico, for the purposes of massively

parallel oligo synthesis and single-cell analytics, respectively. The other authors declare that no competing interests exist.

## Funding

| Funder | Grant reference number | Author |
|---|---|---|
| National Cancer Institute | P01 CA013106 | Scott W Lowe |
| National Cancer Institute | R01 CA233944 | Scott W Lowe |
| National Institute of General Medical Sciences | 1K99GM140265-01 | Yadira M Soto-Feliciano |
| National Cancer Institute | 1F32CA257103 | Changyu Zhu |
| American Cancer Society | PF-14-066-01-TBE | John P Morris |
| Helmholtz foundation | VH-NG-1114 | Darjus F Tschaharganeh |
| National Cancer Institute | P30 CA008748 | Scott W Lowe |

The funders had no role in study design, data collection and interpretation, or the decision to submit the work for publication.

## Author contributions

Changyu Zhu, Yadira M Soto-Feliciano, Conceptualization, Data curation, Formal analysis, Validation, Investigation, Visualization, Methodology, Writing – original draft, Writing – review and editing; John P Morris, Conceptualization, Data curation, Formal analysis, Validation, Investigation, Methodology, Writing – original draft, Writing – review and editing; Chun-Hao Huang, Conceptualization, Data curation, Formal analysis, Validation, Investigation, Methodology, Writing – original draft; Richard P Koche, Yu-jui Ho, Software, Formal analysis, Visualization, Methodology; Ana Banito, Data curation, Methodology; Chun-Wei Chen, Formal analysis; Aditya Shroff, Sha Tian, Geulah Livshits, Chi-Chao Chen, Myles Fennell, Scott A Armstrong, Methodology; C David Allis, Supervision, Writing – review and editing; Darjus F Tschaharganeh, Conceptualization, Data curation, Formal analysis, Supervision, Validation, Investigation, Methodology, Writing – original draft, Writing – review and editing; Scott W Lowe, Conceptualization, Resources, Supervision, Funding acquisition, Investigation, Writing – original draft, Project administration, Writing – review and editing

## Author ORCIDs

Changyu Zhu http://orcid.org/0000-0003-3583-3638
Yadira M Soto-Feliciano http://orcid.org/0000-0002-8523-7917
Richard P Koche http://orcid.org/0000-0002-6820-5083
Ana Banito http://orcid.org/0000-0003-2188-0003
Chun-Wei Chen http://orcid.org/0000-0002-8737-6830
Scott A Armstrong http://orcid.org/0000-0002-9099-4728
Scott W Lowe http://orcid.org/0000-0002-5284-9650

## Ethics

All animal experiments were approved by the MSKCC Institutional Animal Care and Use Committee (protocol 11-06-011). Animals were monitored for signs of ill-health by veterinary staff at the Research Animal Resource Center (RARC) at MSKCC and efforts were made to minimize suffering.

## Decision letter and Author response

Decision letter https://doi.org/10.7554/eLife.80854.sa1
Author response https://doi.org/10.7554/eLife.80854.sa2

# Additional files

## Supplementary files

• Supplementary file 1. Tables displaying the sequences of single guide RNA (sgRNA), shRNA, qPCR primers, and chromatin immunoprecipitation (ChIP)-qPCR primers used in the studies of this manuscript.

• MDAR checklist

## Data availability

Source files of all original gels and Western Blots were provided for the following figures: Figure 1—figure supplement 2B; Figure 4—figure supplement 1A, C, D, E; Figure 5—figure supplement 2B, F, G. RNA sequencing and ChIP sequencing data files that support the findings of this study have been deposited in the Gene Expression Omnibus under the accession code GSE85055, as well as in the Dryad digital repository (doi:10.5061/dryad.7pvmcvdwm; doi:10.5061/dryad.f1vhhmh0h). Sequences of sgRNAs, shRNAs, and primers used in this manuscript are included in the Supplementary File 1.

The following datasets were generated:

| Author(s) | Year | Dataset title | Dataset URL | Database and Identifier |
|---|---|---|---|---|
| Soto-Feliciano MY, Zhu C, Morris JP, Huang C-H, Koche RP, Y-J Ho, Banito A, Chen C-W, Shroff A, Tian S, Livshits G, Chen C-C, Fennell M, Armstrong SA, Allis CD, Tschaharganeh DF, Lowe SW | 2022 | Mll3 suppresses tumorigenesis by activating the Ink4a/Arf locus | https://doi.org/ 10.5061/dryad. 7pvmcvdwm | Dryad Digital Repository, 10.5061/dryad.7pvmcvdwm |
| Soto-Feliciano MY, Zhu C, Morris JP, Huang C-H, Roche RP, Y-J Ho, Banito A, Chen C-W, Shroff A, Tian S, Livshits G, Chen C-C, Fennell M, Armstrong SA, Allis CD, Tschaharganeh DF, Lowe SW | 2022 | MLL3 ChIP sequencing in murine and human HCC cells | https://doi.org/ 10.5061/dryad. f1vhhmh0h | Dryad Digital Repository, 10.5061/dryad.f1vhhmh0h |
| Lowe SW | 2017 | Mll3 suppresses tumorigenesis by activating the Ink4a/Arf locus | https://www.ncbi. nlm.nih.gov/geo/ query/acc.cgi?acc= GSE85055 | NCBI Gene Expression Omnibus, GSE85055 |

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

# Appendix 1

## Appendix 1—key resources table

| Reagent type (species) or resource | Designation | Source or reference | Identifiers | Additional information |
|---|---|---|---|---|
| Strain and strain background (*M. musculus*) | Wild-type C57BL/6 J | The Jackson Laboratory | Stock #000664 | |
| Cell line (*Homo-sapiens*) | HLE HCC cell line | JCRB Cell Bank | JCRB0404 | |
| Cell line (*M. musculus*) | *Myc*; sg*Trp53* HCC cell lines | This paper | NA | Three independent cell lines derived from different mice were used as biological replicates |
| Cell line (*M. musculus*) | *Myc*; sg*Kmt2c* HCC cell lines | This paper | NA | Three independent cell lines derived from different mice were used as biological replicates |
| Cell line (*M. musculus*) | *Myc*; sg*Axin1* HCC cell lines | This paper | NA | Three independent cell lines derived from different mice were used as biological replicates |
| Cell line (*M. musculus*) | TRE-sh*Kmt2c*.1 liver progenitor line | This paper | NA | |
| Cell line (*M. musculus*) | TRE-sh*Kmt2c*.2 liver progenitor line | This paper | NA | |
| Antibody | Anti-MLL3/4 (Rabbit polyclonal) | *Dorighi et al., 2017*; PMID:28483418 | NA | ChIP-seq (1:500) |
| Recombinant DNA reagent | pT3-*Myc* | Addgene | #92046 | |
| Recombinant DNA reagent | pT3-*Ctnnb1 N90* | Addgene | #31785 | |
| Recombinant DNA reagent | PX330-Cas9-U6-sgRNA | Addgene | #42230 | |
| Recombinant DNA reagent | CMV-SB13 | *Huang et al., 2014*; PMID:25128497 | NA | |
| Recombinant DNA reagent | pT3-EF1a-GFP-miRE | *Huang et al., 2014*; PMID:25128497 | NA | |

