## [Editor Report]

This paper convincingly shows that MLL3 regulates the CDKN2A tumor suppressor in MYC-driven liver cancers. The use of in vivo models and epigenomic analysis made the findings particularly robust. This work significantly advances our understanding of the function of MLL3 in cancer.

---

## [Decision Letter]

**Decision letter after peer review:**

Thank you for submitting your article "MLL3 regulates the *CDKN2A* tumor suppressor locus in liver cancer" for consideration by *eLife*. Your article has been reviewed by 3 peer reviewers, including Hao Zhu as the Reviewing Editor and Reviewer #1, and the evaluation has been overseen by Erica Golemis as the Senior Editor. The following individual involved in the review of your submission has agreed to reveal their identity: Jeffrey A Magee (Reviewer #2).

Essential revisions:

There are several suggestions or clarifications from each reviewer that need to be addressed; Please address all of the comments from the reviewers.

*Reviewer #1 (Recommendations for the authors):*

1. How can we tell that the ChIP-seq signal is not coming entirely from MLL4 in the MYC; sgMLL3 tumors? if the signal is partially or entirely from MLL4, then it is hard to know what amount of binding is contributed by MLL3. Moreover, there could be concomitant up-regulation of MLL4 when MLL3 is deleted.

2. Figure 1C should have some statistical analysis. The numbers of mice used here are low, so the stats should be included.

3. Figure 4 D, F, and G would benefit from an increased number of replicates.

4. Figure S1D is missing "T" for transposons.

*Reviewer #2 (Recommendations for the authors):*

1. There are no statistical arguments to support the claim that MLL3 and CDKN2A mutations are mutually exclusive in HCC or any of the other cancers that they analyzed. When each mutation is relatively rare, is it that surprising that they do not co-occur? The authors should support this argument with a more rigorous analysis or remove it.

2. The authors argue that MLL3 binds the CDKN2A promoter, but the ChIP-seq data also show peaks within the gene body. Can they clarify how they distinguish direct interactions between MLL3 and promoter elements from indirect interactions that might occur when enhancers loop and engage the promoter?

3. The authors should show a Western blot demonstrating specific loss of MLL3 expression in the sgMll3 lines, given that the antibody used for the ChIP-seq studies recognizes both MLL3 and MLL4.

4. Figures 2C and 3B, C both need clarification. For Figure 2C, what do the values in the heatmap reflect? For Figure 3C, how did they generate the gene set for the GSEA, and what are these comparisons meant to prove? This point did not come through clearly in the manuscript.

5. The study focuses entirely on MYC-driven HCC, with the exception of the CRISPRa experiments performed in p53/Tert mutant HLE cells. Can the authors comment on whether the MLL3/CDKN2A interaction generalizes to non-MYC-driven HCC?

*Reviewer #3 (Recommendations for the authors):*

The authors of the manuscript describe an interesting and novel mechanism by which MLL3 regulates the CDKN2A tumor suppressor locus in MYC amplification-dependent liver cancer.

Overall, the manuscript is well written, organized, and focused on an interesting topic, and with data presented supports the authors' claims. Nevertheless, there are a few concerns that remain in relating to the study, and therefore, need to be addressed further for publication in the *eLife* journal.

Specific comments:

1. While the mouse genetic model manipulating the overexpression (OE) of oncogene and knockout of tumor suppressor gene (TSG) is an elegant design to show the combined effects of MYC-OE and MLL3-loss in liver tumorigenesis, it might not be a straightforward idea to use Myc; sgp53 as a control, especially in the ChIP-seq study and the downstream analyses. Because p53 has its own distinct set of targets that may cooperate with MYC and p53 negatively regulates Ink4a and Arf expression, the authors should explain more clearly the rationale for choosing Myc; sgp53, rather than Myc; sgChr8 (only overexpress Myc but does not silence any targets), as a control in the comparison with Myc; sgMll3.

2. Although the authors provide a potential mechanism for MLL3's tumor suppressive role in MYC-driven tumorigenesis, the clinical relevance is shown in Figure 1A may only fit their hypothesis into a small fraction of human HCC patients due to the relatively low mutation rate of MLL3 (4%) compared to the high MYC amplification/gain mutation rate (41%). In addition, it would be interesting to assess the clinical relevance between amplification/gain mutations of MYC and truncating mutations/deep deletions of MLL3 or CDKN2A in human cancers shown in figures S4C and S4D. This will strengthen their hypothesis that loss of MLL3 or CDKN2A impairs MYC-induced apoptosis, but instead, contributes to MYC-induced tumorigenesis in vivo.

3. The supplementary experiments described in lines 179 – 188 to "demonstrate a direct transcriptional effect of MLL3 on the Cdkn2a locus" were not very clear and somehow confusing to the reviewer the first time reading and it was unclear why the authors involved p53 and Axin1 targeting in this set of experiments.

4. The novel findings of this manuscript would be: (i) MLL3 regulates tumor suppressor gene expression via modulating their promoter chromatin landscapes (i.e. H3K4me3), and (ii) loss of normal function of MLL3 or the downstream effector CDKN2A may impair MYC-induced apoptosis, and in turn, lead to MYC-induced tumorigenesis. Thus, it would be better to introduce more background regarding the current knowledge of (i) MLL3's roles in the regulation of enhancer-associated H3K4me1 modulation and somehow unclear but detectable function in promoter-associated H3K4me3 regulation, and (ii) the cellular content-dependent regulation of MYC-induced apoptosis or tumorigenesis. The current introduction is too short.

---

## [Author Response]

Essential revisions:There are several suggestions or clarifications from each reviewer that need to be addressed; Please address all of the comments from the reviewers.Reviewer #1 (Recommendations for the authors):1. How can we tell that the ChIP-seq signal is not coming entirely from MLL4 in the MYC; sgMLL3 tumors? if the signal is partially or entirely from MLL4, then it is hard to know what amount of binding is contributed by MLL3. Moreover, there could be concomitant up-regulation of MLL4 when MLL3 is deleted.

We agree that the interpretation may be limited due to the fact that the antibody used for ChIP recognizes both MLL3 and MLL4. Unfortunately, this was the only antibody we found effective for the murine systems after years of trying. Thus, in comparing *Myc*; sg*Kmt2c* tumors, with *Myc*; *sgTrp53* tumors, the unchanged peak signals in some areas of the genome are most likely due to MLL4 binding. It is important to note that this overlapping specificity highlights an exclusive relationship between MLL3 and targets such as *Cdkn2a*, as our RNA-seq data demonstrate that there was no compensatory change in *Kmt2d* (Mll4) expression in *Kmt2c* targeted *cells* (Figure 3—figure supplement 1A), where we observe a dramatic reduction of peaks associated with *Cdkn2a* locus by ChIP-seq. From these results, we conclude that the *Cdkn2a* locus is a tumor suppressive genomic target of MLL3. We acknowledge the limitations of the antibody and how this effects our interpretations in the revised text.

2. Figure 1C should have some statistical analysis. The numbers of mice used here are low, so the stats should be included.

We now include the statistics (the statistical method and *P*-values) in all relevant figures.

3. Figure 4 D, F, and G would benefit from an increased number of replicates.

We repeated the experiments with additional biological replicates (n = 4). Importantly, we also tested an additional sg*Kmt2c* for CRISPRa to demonstrate the effects are not due to one specific guide RNA.

4. Figure S1D is missing "T" for transposons.

We re-made the figure to correct this omission.

Reviewer #2 (Recommendations for the authors):1. There are no statistical arguments to support the claim that MLL3 and CDKN2A mutations are mutually exclusive in HCC or any of the other cancers that they analyzed. When each mutation is relatively rare, is it that surprising that they do not co-occur? The authors should support this argument with a more rigorous analysis or remove it.

We agree with the reviewer that the mutually exclusive mutation patterns between *KMT2C* and *CDKN2A* in human cancers are not statistically supported, mainly due to the low mutation frequencies of one or both genes. Thus, we removed the majority of the mutation analyses. The updated oncoprint in HCC is modified to include *MYC* and all the *P*-values of the comparisons of interest.

2. The authors argue that MLL3 binds the CDKN2A promoter, but the ChIP-seq data also show peaks within the gene body. Can they clarify how they distinguish direct interactions between MLL3 and promoter elements from indirect interactions that might occur when enhancers loop and engage the promoter?

We agree with the reviewer that we cannot rule out other possible promoter-engaging mechanisms, such as via MLL3 binding within the gene body and/or looping between promoters and enhancers. To address some of these concerns, we also performed ChIP-seq for several histone modifications that have been widely used to identify different cis-acting elements in the genome and their transcriptional state (H3K4me1 – enhancer, H3K4me3 – promoter, H3K27ac – active enhancer or promoter). We also modified the relevant text in the Results and in the Discussion to incorporate the reviewer’s point.

3. The authors should show a Western blot demonstrating specific loss of MLL3 expression in the sgMll3 lines, given that the antibody used for the ChIP-seq studies recognizes both MLL3 and MLL4.

We agree with the reviewer that a Western blot result would further support our conclusion. To address this, we tried several anti-mouse MLL3 antibodies, including one from Cell Signaling Technology (#53641) and another non-commercially available one that worked for ChIP-seq. Unfortunately, none of these attempts yielded specific bands with correct molecular weight in the immunoblots.

4. Figures 2C and 3B, C both need clarification. For Figure 2C, what do the values in the heatmap reflect? For Figure 3C, how did they generate the gene set for the GSEA, and what are these comparisons meant to prove? This point did not come through clearly in the manuscript.

We thank the reviewer for pointing out the lack of clarity, and we have updated the text in the Results and the Methods to provide the information requested.

For Figure 2C, the values in the heatmap are the Z-scores of the normalized counts for the peak signals of histone modifications in ChIP-seq data. For GSEA in Figure 3, transcriptional signatures were derived based on significantly changed genes (adjusted *P* <0.05; absolute fold change >2) in RNA-seq on mouse HCC cell lines (*Myc*; sg*Kmt2c* vs *Myc*; sg*Trp53*), and in human HCCs with mutations in *KMT2C* vs. *TP53* (adjusted *P* <0.05; absolute fold change >2). Those signatures were then compared to the transcriptional results on human HCCs with genomic inactivation of *CDKN2A* vs *TP53* in Figure 3B and 3C. We updated Figure 3 to add more relevant details.

5. The study focuses entirely on MYC-driven HCC, with the exception of the CRISPRa experiments performed in p53/Tert mutant HLE cells. Can the authors comment on whether the MLL3/CDKN2A interaction generalizes to non-MYC-driven HCC?

Our study primarily used *Myc* as the driving oncogene for two reasons: first, MLL3 was nominated as a tumor suppressor gene whose loss cooperates with *Myc* oncogene in an in vivo HCC screen (new Figure 1—figure supplement 1); second, in human HCCs, *KMT2C* mutations and deletions co-occur with *MYC* gains and amplification.

We agree with the reviewer that it is of interest to know whether the results are oncogene independent. Thus, we performed the analogous experiment by HTVI of transposon containing the constitutively active *Ctnnb1*. However, we did not observe oncogenic cooperation between *Ctnnb1* activation and MLL3 loss; no mice developed liver tumors by the experimental endpoint (5 months post HTVI, Figure 1—figure supplement 3). Additionally, analysis of human HCCs showed no significant co-occurrence between *CTNNB1* and *KMT2C* alterations (Figure 1A). These results suggests that, similar to other epigenetic regulators, the tumor-suppressive function of MLL3 is likely oncogene- and context dependent. Recent studies support the notion that chromatin context can favor certain oncogenic alterations and we speculate in the discussion that this may also be true for MLL3. We have updated the text to reflect the context-specificity of MLL3 as a tumor suppressor.

Reviewer #3 (Recommendations for the authors):The authors of the manuscript describe an interesting and novel mechanism by which MLL3 regulates the CDKN2A tumor suppressor locus in MYC amplification-dependent liver cancer.Overall, the manuscript is well written, organized, and focused on an interesting topic, and with data presented supports the authors' claims. Nevertheless, there are a few concerns that remain in relating to the study, and therefore, need to be addressed further for publication in the eLife journal.Specific comments:1. While the mouse genetic model manipulating the overexpression (OE) of oncogene and knockout of tumor suppressor gene (TSG) is an elegant design to show the combined effects of MYC-OE and MLL3-loss in liver tumorigenesis, it might not be a straightforward idea to use Myc; sgp53 as a control, especially in the ChIP-seq study and the downstream analyses. Because p53 has its own distinct set of targets that may cooperate with MYC and p53 negatively regulates Ink4a and Arf expression, the authors should explain more clearly the rationale for choosing Myc; sgp53, rather than Myc; sgChr8 (only overexpress Myc but does not silence any targets), as a control in the comparison with Myc; sgMll3.

We agree with the reviewer that using *Myc*; sg*Trp53* cells as controls has downsides, due to p53specific tumor suppressive functions. In theory, HCC cell lines with only *Myc* activation would have been ideal. However, as shown in Figure 1, *Myc* activation alone (*Myc*; sg*Chrom8*) does not yield liver tumors from which we could eventually derive cell lines and, even had they occurred, there would likely be selected secondary events.

We also agree that silencing p53 may result in compensatory upregulation of Ink4a and Arf. However, we performed a number of experiments to validate the ability of MLL3 to directly regulate *Cdkn2a* in p53 null and WT cells (Figure 4—figure supplement 1). We found that targeting *Kmt2c* with 2 sgRNAs reduces p16^Ink4a^ and p19^Arf^ levels in *Myc*; sh*Trp53* cells (Figure 4—figure supplement 1D), and *Kmt2c* shRNA knockdown reduces Ink4a and Arf transcripts in *Myc*; sg*Axin1* cells where p53 was not altered (Figure 4—figure supplement 1F). All of these observations indicate that the regulation of the *Cdkn2a* locus is MLL3-dependent rather than the result of p53 loss. Lastly, new Figure 4—figure supplement 1G shows that MLL3 binding peaks at *Cdkn2a* locus also exist in *Myc*; sg*Axin1* cells, confirming that our results and conclusions are not exclusive to p53 inactivated cells. We acknowledge these potential confounding issues with compensatory activation of *Cdkn2a* and these confirmatory experiments in the revised text.

2. Although the authors provide a potential mechanism for MLL3's tumor suppressive role in MYC-driven tumorigenesis, the clinical relevance is shown in Figure 1A may only fit their hypothesis into a small fraction of human HCC patients due to the relatively low mutation rate of MLL3 (4%) compared to the high MYC amplification/gain mutation rate (41%). In addition, it would be interesting to assess the clinical relevance between amplification/gain mutations of MYC and truncating mutations/deep deletions of MLL3 or CDKN2A in human cancers shown in figures S4C and S4D. This will strengthen their hypothesis that loss of MLL3 or CDKN2A impairs MYC-induced apoptosis, but instead, contributes to MYC-induced tumorigenesis in vivo.

We thank the reviewer for these constructive suggestions. We recognize the limitations of our previous analyses of *KMT2C* and *CDKN2A* mutations in different cancer types. Because *MYC* is a recurrent oncogene in some but not all cancer types, we focused the analysis to show statistically significant cooccurrences between *MYC* and *KMT2C*/*CDKN2A* mutations in HCC, and removed the less relevant oncoprints.

3. The supplementary experiments described in lines 179 – 188 to "demonstrate a direct transcriptional effect of MLL3 on the Cdkn2a locus" were not very clear and somehow confusing to the reviewer the first time reading and it was unclear why the authors involved p53 and Axin1 targeting in this set of experiments.

We apologize for the confusion, and thank the reviewer for pointing it out. As referred to in our response to the first point, it is documented that p53 loss could result in activation of Ink4a and Arf. To rule out such a possibility, we targeted *Kmt2c* by CRISPR in *Myc*; sh*Trp53* cells to demonstrate that MLL3 controls p16^Ink4a^ and p19^Arf^ expression in p53-suppressed tumor cells. Moreover, both shRNA knockdown of *Kmt2c* and ChIP-seq on MLL3 in *Myc*; sg*Axin1* cells further show that the regulation of *Cdkn2a* by MLL3 is independent of p53 status. We have modified the text to more accurately and clearly reflect our results.

4. The novel findings of this manuscript would be: (i) MLL3 regulates tumor suppressor gene expression via modulating their promoter chromatin landscapes (i.e. H3K4me3), and (ii) loss of normal function of MLL3 or the downstream effector CDKN2A may impair MYC-induced apoptosis, and in turn, lead to MYC-induced tumorigenesis. Thus, it would be better to introduce more background regarding the current knowledge of (i) MLL3's roles in the regulation of enhancer-associated H3K4me1 modulation and somehow unclear but detectable function in promoter-associated H3K4me3 regulation, and (ii) the cellular content-dependent regulation of MYC-induced apoptosis or tumorigenesis. The current introduction is too short.

We thank the reviewer for this suggestion. We have revised the Introduction to weave in these two points.